# Extreme lowering of deglacial seawater radiocarbon recorded by both epifaunal and infaunal benthic foraminifera in a wood-dated sediment core

**Authors:** Patrick A. Rafter[1]*, Juan-Carlos Herguera[2], and John R. Southon[1]

**Affiliations:**

[1]Department of Earth System Science, University of California, Irvine, CA, USA

[2]Centro de Investigación Cientifica y Educación Superior de Ensenada

*Correspondence to: prafter@uci.edu

**Key Points**

Carbon Cycle

Climate

Ice ages

**Abstract:**

For over a decade, oceanographers have debated the interpretation and reliability of sediment microfossil records indicating extremely low seawater radiocarbon ($^{14}$C) during the last deglaciation—observations that suggest a major disruption in marine carbon cycling coincident with rising atmospheric $CO_2$ concentrations. Possible flaws in these records include poor age model controls, utilization of mixed, infaunal foraminifera species, and bioturbation. We have addressed these concerns using a glacial-interglacial record of epifaunal benthic foraminifera $^{14}$C on an ideal sedimentary age model (wood calibrated to atmosphere $^{14}$C). Our results affirm—with important caveats—the fidelity of these microfossil archives and confirm previous observations of highly depleted seawater $^{14}$C at intermediate depths in the deglacial northeast Pacific.

## 1.0 Introduction

Given modern carbon cycle perturbations (Keeling, 1960), it is critical to understand the drivers of natural atmospheric carbon dioxide ($CO_2$) variability. A prime example of this 'natural' atmospheric $CO_2$ variability is the increase that occurs at the end of each late-Pleistocene ice age (Figure 1) (Petit et al., 1999). The ocean's ability to store and release $CO_2$ makes it a likely driver of past changes in this important greenhouse gas (Broecker, 1982).

A valuable tool in the effort to characterize the marine carbon cycle over the most recent of these intervals is the $^{14}$C content of benthic and planktic foraminifera tests (Broecker et al., 1988), which are assumed to reflect the $^{14}$C content of dissolved inorganic carbon (DIC) in the waters in which they grew. This tracer provides a geochemical "clock" with a predictable decay but $^{14}$C is also affected by a variety of other processes, including the time since the water mass exchanged $CO_2$ with the atmosphere, the degree of this exchange, variations in the atmospheric

concentration of [14]C at the time of exchange (Figure 1), as well as the contribution of
[14]C-depleted carbon via mixing and / or other carbon sources (e.g., seafloor
volcanism (Ronge et al., 2016)).
We can relate seawater [14]C content to modern ocean conditions by using delta
notation or $\Delta^{14}$C (Figure 1), which corrects for [14]C decay:
$\Delta^{14}$C = e^(-[14]C age/8033) / e^(-Calendar Age/8267) – 1          (1)
(Equation (1) is multiplied by 1000 to give units of per mil [‰]. The [14]C age
Calendar Age is given in years before 1950 or "before present" (BP).)
The available benthic foraminifera $\Delta^{14}$C records paint a complicated picture of
glacial to interglacial seawater [14]C content. For example, a record of benthic
foraminifera $\Delta^{14}$C from the intermediate depth subtropical eastern North Pacific
(Lindsay et al., 2015; Marchitto et al., 2007) shows $\Delta^{14}$C depleted relative to the
atmosphere by >400‰ during the deglaciation (from ≈19-to-11,000 years BP; see
Figure 1). Later work showed benthic foraminifera with similar or even lower $\Delta^{14}$C
values during the deglaciation in other parts of the intermediate depth ocean (≈500-
1000 m), such as the 617 m deep Eastern Equatorial Pacific (Stott et al., 2009) and
the 596-820 m deep Arabian Sea (Bryan et al., 2010). Given that the lowest observed
modern intermediate-depth seawater $\Delta^{14}$C is about -300‰ (or only ≈300‰ lower
than the atmosphere) (Key et al., 2004), the low benthic foraminifera $\Delta^{14}$C / old [14]C
ages suggest much lower $\Delta^{14}$C  and older seawater DIC [14]C ages during the
deglaciation.
A leading explanation of these low intermediate depth $\Delta^{14}$C values involves the
storage of carbon in an isolated deep-sea reservoir during the glacial period
followed by the rapid flushing of this low $\Delta^{14}$C / old [14]C aged carbon through the
intermediate-depth ocean during the deglaciation—a deep-sea carbon flush that
also explains the observed elevation of  atmospheric $CO_2$ concentrations and
lowering of atmospheric $CO_2$ [14]C content (Marchitto et al., 2007). This interpretation
is qualitatively supported by observations of lower deep-sea dissolved oxygen
concentrations before the deglaciation (Jaccard et al., 2016; Jaccard and Galbraith,
80  2011).

The ocean carbon flushing hypothesis predicts that deep-sea $\Delta^{14}$C during the glacial
period will be lower than the extreme $\Delta^{14}$C lowering of the intermediate-depth $\Delta^{14}$C
during the deglaciation (Figure 1) because of mixing with shallower waters with
higher $\Delta^{14}$C. However, while deglacial $\Delta^{14}$C as low or lower than in Figure 1 is
observed in some deep-sea waters during the glacial period (Sikes et al., 2000;
Skinner et al., 2010; Keigwin and Lehman, 2015) and intermediate-depth waters
(Burke and Robinson, 2012)—observations that are consistent with the flushing
hypothesis—it is not clear how these low $\Delta^{14}$C signals are not mixed away en route
to the lower latitudes (Hain et al., 2011). Additionally, the lower $\Delta^{14}$C in Figure 1 is
not observed at all intermediate depth sites during the deglaciation (De Pol-Holz et
al., 2010; Rose et al., 2010). Furthermore, the extreme $\Delta^{14}$C lowering observed in
intermediate-depth benthic foraminifera during the deglaciation does not appear to
be quantitatively consistent with an isolated deep-sea reservoir (Hain et al., 2011).
The inconsistency of the available $\Delta^{14}$C records is compounded by assumptions
about the reliability of the foraminifera archive as a recorder of seawater DIC $^{14}$C.
For example, an important assumption when using planktic foraminifera is that the
depth of calcification does not vary based on modern observations (e.g., (Field,
2004)). The use of benthic foraminifera seemingly circumvents this problem, and
those that live at the sediment-water interface ("epifaunal") have been
demonstrated to record seawater carbon chemistry (Keigwin, 2002; Roach et al.,
2013). However, the abundance of epifaunal benthic foraminifera is typically low
relative to benthic species that abide within the sediment ("infaunal"). Rather than
recording seawater $^{14}$C content directly, the infaunal species provide a record of
sediment pore water carbon chemistry, which may or may not reflect bottom water
conditions.
A further complication to published benthic foraminifera $\Delta^{14}$C observations is that
both the epifaunal and infaunal species are typically rare in sediments, leading to
the common use of mixed benthic species. The mixed species approach has led, in
some rare cases, to anomalously low $\Delta^{14}$C values / old $^{14}$C ages by inclusion of
anomalously depleted $^{14}$C *Pyrgo* spp. (Magana et al., 2010)—an anomaly that may
not be a global phenomenon (Thornalley et al., 2015). While mono-species epifaunal
benthic foraminifera $^{14}$C measurements exist (Thornalley et al., 2011, 2015; Voelker
et al., 1998), we are unaware of any continuous glacial-interglacial records of mono-
species epifaunal foraminifera $^{14}$C content. (One study used mixed planispiral
species, whose morphology predicts an epifaunal habitat (Galbraith et al., 2007).)
An additional influence on benthic foraminifera $\Delta^{14}$C is bioturbation (Keigwin and
Guilderson, 2009), which is infrequently quantified, even though it can dramatically
affect the observed $^{14}$C age (Costa et al., 2018). The doubts raised by the above
complications are amplified by converting the benthic foraminifera $^{14}$C age to $\Delta^{14}$C,
which requires the user to assign a calendar age to the sediment.
Finally, constraining the age model of sediment cores typically relies upon several
assumptions. For example, planktic foraminifera $^{14}$C is commonly used to identify
the calendar age of sedimentary material, although this requires assumptions about
the depth habitat of the planktic foraminifera and the 'reservoir age' of the surface
waters (the offset between atmosphere and ocean $^{14}$C). Other means for
determining the calendar age involve tying temporal variability to other
paleoclimate/paleoceanographic records (Marchitto et al., 2007; Stott et al., 2009).
In rare instances, the $^{14}$C of wood from terrestrial plants provides a direct recording
of atmospheric $^{14}$C, which is well-dated and provides an excellent sedimentary age
model (Broecker, 2004; Zhao and Keigwin, 2018), although this work provides some
recommendations for utilizing this technique (see below). For our understanding of
past and future carbon cycling processes, it is essential that we thoroughly explore
these influences and build confidence in these sediment proxy records.
Here, we provide a test of the fidelity of the benthic foraminifera $\Delta^{14}$C proxy using
$^{14}$C measurement of benthic foraminifera species from two sediment cores near the
mouth of the Gulf of California (white diamond in Figure 2). These sediment cores
are unusual in that both epifaunal and infaunal benthic foraminifera microfossils are
plentiful and allow us a unique opportunity to test the fidelity of the benthic
foraminifera $\Delta^{14}$C proxy. The foraminiferal abundance were quantified to account
for bioturbation and the age model is calibrated to the well-constrained
atmospheric $^{14}$C record (Reimer et al., 2013) via wood found alongside the
foraminifera. These cores (from hereon, the 'Gulf' sites) allow us to present glacial-
interglacial $^{14}$C measurements produced from 4 benthic foraminifera, including the
preferred epifaunal species *Planulina ariminensis* (Keigwin, 2002). The Gulf core
sites are bathed in the subsurface, northward flowing Mexican Coastal Current (MCC
in Figure 2), which are the source of the California Undercurrent (Gómez-Valdivia et
al., 2015)—waters that also bathe the well known sites on the Pacific margin of Baja
California shown in Figure 1 (from hereon, the 'California Undercurrent' sites). This
shared seawater source gives the expectation of similar $\Delta^{14}$C signal at both
sedimentary locations—an expectation that we exploit to examine the potential for
diagenetic alteration of the benthic foraminifera $\Delta^{14}$C observations relative to
sedimentation rates, which are significantly lower at the Gulf sites ($\approx$2 to 5 cm kyr$^{-1}$;
our study) relative to the Undercurrent sites (>25 cm kyr$^{-1}$; (Lindsay et al., 2015;
Marchitto et al., 2007)) (where 'kyr' is 1000 years). These and other hydrological,
geochemical, and diagenetic influences on benthic foraminifera $\Delta^{14}$C are examined
below with the goal of answering an important question: are these benthic
foraminifera $\Delta^{14}$C records recording an extreme lowering of seawater $\Delta^{14}$C during
the deglaciation?
**2.0 Materials and Methods**
Sediment from Gulf of California sites LPAZ-21P (22.9°N, 109.5°W; 625 m) and
ET97-7T (22.9°N, 109.5°W; 640 m) (white diamond in Figure 2; Table 1) was
washed using de-ionized water in a 63-µm sieve. Foraminifera abundance estimates
of *Planulina ariminensis* (benthic; epifaunal species), *Uvigerina peregrina* (benthic;
shallow infaunal species), *Trifarina bradyi* (benthic; deep infaunal species), mixed
*Bolivina* (benthic; deep infaunal species), and *Globogerina bulloides* (planktic
species) were made after quantitatively dividing the >150 µm fraction of each
sample using a Green Geological aluminum microsplitter. These estimates were
made for all samples from core LPAZ-21P and select samples from core ET97-7T.
Preliminary work measured the $^{14}$C age of mixed benthic species from the ET97-7T
core site and although the species abundance was not quantified, they primarily
included *Planulina* spp., *Uvigerina* spp., and *Trifarina* spp.
**2.1 Radiocarbon measurements**
Monospecies foraminifera and wood were selected for $^{14}$C analysis from the >250
μm fraction from both Gulf sediment cores. Each foraminifera sample was sonicated
in methanol (≈1 minute) to release detrital carbonates trapped within open
microfossil chambers. At least 10% of each sample was dissolved using HCl to
remove secondary calcite (precipitated post-deposition), though in-house tests with
and without this pretreatment yielded identical results for these core sites. Wood
fragments from the >250 μm fraction were prepared using standard acid-base-acid
treatments.
Samples were graphitized following (Santos et al., 2007) and analyzed at the Keck
Carbon Cycle Accelerator Mass Spectrometry (KCCAMS) laboratory at University of
California, Irvine (Southon et al., 2004). We report radiocarbon as $\Delta^{14}$C in units of
per mil [‰] (see equation (1) above), which is corrected for decay based on its age
normalized to 1950, according to convention (Stuiver and Polach, 1977). Analysis of
a sedimentary standard (FIRI-C) alongside measurements indicates a combined
sample preparation and measurement $^{14}$C age error ranging from ±50 years for a
full size sample (≈0.7 mg of C) to ±500 years for very small samples (<0.1 mg of C).
Because of the similar location of the sites near the mouth of the Gulf of California,
we combined the $^{14}$C measurements from both cores.
**2.2 Oxygen and carbon stable isotopic measurements**
The $^{18}$O/$^{16}$O and $^{13}$C/$^{12}$C of benthic foraminifera was measured using a Kiel IV
Carbonate Device coupled to a Delta XP isotope ratio mass spectrometer at the
University of California, Irvine. Isotopic ratios are reported in delta notation, where:
$\delta^{13}$C = ($^{13}$C/$^{12}$C$_{sample}$ / $^{13}$C/$^{12}$C$_{standard}$ − 1) and $\delta^{18}$O = ($^{18}$O/$^{16}$O$_{sample}$ / $^{18}$O/$^{16}$O$_{standard}$ -1).
Each was multiplied by 1000 to give units of "per mil". The standard for both
measurements is VPDB.
**2.3 Age model construction for Gulf of California sediment cores**
The age model for LPAZ-21P (between 30,000-to-12,100 years Before Physics or
"BP," where BP is 1950) is constrained by 13 microscopic wood fragments
calibrated to calendar ages using CALIB7.1 (Stuiver et al., 2017) with the IntCal13
atmospheric $^{14}$C dataset (Reimer et al., 2013) (squares in Figure 3A and 3C). Five
wood measurements from LPAZ-21P did not pass our test for use as an age model
constraint (upside-down triangles in Figure 3C and see text below). All LPAZ-21P
depths shallower than 63 cm are notably darker, changing from light to very dark
brown over a depth interval of ≈2 cm. The onset of this change is constrained to be
younger than 12,100±1,100 years BP (12.1±1.1-kyr BP) by a calibrated wood $^{14}$C
age (see Appendix). There was a lack of suitable wood in LPAZ-21P in Holocene-
aged sediments and our age models for this interval are constrained using *U.*
*peregrina* $^{14}$C ages (circles in Figure 3A), corrected for a modern reservoir age of
1240 years based on nearby seawater DIC $^{14}$C age observations at 600 m (Key et al.,
2004) and converted to calendar ages using CALIB7.1 (Stuiver et al., 2017). These
Holocene $^{14}$C ages are not tied to foraminifera abundance maxima and hence the
Holocene calendar ages should be considered preliminary. The youngest calendar
age for LPAZ-21P was 5.3-kyr BP, suggesting piston core over-penetration during
sediment coring. Samples younger than the LPAZ-21P coretop were obtained from
the LPAZ-21PG core, whose age model was constrained identical to the Holocene-
aged sediments of LPAZ-21P (see above). The Bayesian age model program BACON
(Blaauw and Christen, 2011) was used to estimate the age and model error between
the age model constraints.
The ET97-7T age model is constrained in three ways: using $^{14}$C ages of 5 pieces of
microscopic wood from 18.9- to 15.3-kyr BP (diamonds in Figure 3A and 3C); using
*U. peregrina* $^{14}$C ages corrected for reservoir age in Holocene-aged sediment; and by
synchronizing the apparently region-wide transition from light to dark-colored
sediments (van Geen et al., 2003) to 12.1-kyr based on the wood-constrained age
from LPAZ-21P ("X"s in Figure 3A and 3D). In lieu of reflectance data to quantify the
brightness of the sediment cores, we present Ca/Al estimated using X-Ray
Fluorescence (see Method below). The reasoning behind using Ca/Al is that this
metric: (1) Normalizes changes in terrestrial Ca input by dividing by Al and (2) Is
sensitive to the abundance of calcium carbonate microfossils. The sudden lowering
of Ca/Al at ≈65 cm in LPAZ-21P and ≈71 cm in ET97-7T is coincident with
decreased abundance of foraminifera and this presumably causes the darkening of
these and other Holocene-aged sediments across the region. Ages between these
constraints were estimated using BACON, as was done for the LPAZ-21P cores.
## 2.4 X-Ray Fluorescence
We estimated the Ca/Al of LPAZ-21P and ET97-7T using an Avaatech XRF core
scanner at the Scripps Institution of Oceanography Sediment Core Repository. The
archived halves of the sediment cores were lightly scraped to expose less oxidized
sedimentary material before analysis. More detailed methods (including software
and signal processing) are identical to those previously described in (Addison et al.,
253  2013).
## 2.5 Wood $^{14}$C age test
Terrestrial plant life must have a younger $^{14}$C age / higher $\Delta^{14}$C than all
contemporaneous foraminifera because of the air-sea difference in $^{14}$C content (e.g.,
see Figure 1) and we used this inherent $^{14}$C age difference to check for
contemporary deposition of the wood and microfossils in Gulf sediments. Fourteen
out of 20 microscopic wood fragment $^{14}$C ages passed the test and include one
interval that may have been influenced by macrofauna consumption and excretion
has a wood $^{14}$C age that is younger than foraminifera (see below).
One wood measurement that spectacularly failed this test came from presumably
mid-to-late-Holocene sediment (i.e., <12-kyr BP aged sediments based on the depth
below seafloor). However this wood yielded a $^{14}$C age of >25-kyr (see upside down
triangles in Figure 3). We explain this remarkable $^{14}$C age difference as the erosion
and deposition of relict wood stored on land before washing to the Gulf during a
rain event. The other wood measurements that failed this test gave $^{14}$C ages typically
within measurement error or were ≈1000 $^{14}$C years older than foraminifera $^{14}$C age.
In total, 5 out of 20 wood $^{14}$C measurements were older than foraminifera in our
sediment cores relative to 1 out of 26 wood $^{14}$C measurements by the only other
study with similar length age model (Zhao and Keigwin, 2018). This difference may
be because faster sedimentation rate of Zhao and Keigwin, (2018) (20-60 cm kyr$^{-1}$)
leads to less bioturbation and a faster burial of the wood alongside foraminifera
microfossils. Otherwise, the difference in rejections could be explained by our
measurement of all wood, whereas (Zhao and Keigwin, 2018) only measured wood
that still retained bark.
In light of this unusual application of calibrated $^{14}$C ages on wood in a marine setting,
it is important to understand the potential errors. We assigned all calibrated wood
ages a ±100 year uncertainty added in quadrature to the measurement and
calibration error to account for possible lag in seafloor deposition. Note that the
*asymmetry* of any errors associated with assuming contemporary growth of wood
and foraminifera must be considered: if we underestimate the time from wood
growth to sediment deposition, the actual calendar age of the sediment would be
*younger* than the calendar age given in this study; hence foram $\Delta^{14}$C values would be
even *lower* than the large depletions shown here (see equation 1 and Results).
Additionally, it is possible that a longer-than-expected time period between wood
growth and sediment deposition could be "masked" by declining atmospheric $^{14}$C
concentrations (Figure 1), allowing the wood $^{14}$C age to pass our test for inclusion in
the age model. These different histories for the wood found in our sediment cores
would mean the calendar age is younger than we have assumed, adjusting our
benthic foraminifera $\Delta^{14}$C values to lower values than reported below. Given these
potential influences on a wood $^{14}$C age-constrained age model, the uncertainty
should primarily include the younger calendar age and not the ±100 year Gaussian
uncertainty we assume. However, without a more exhaustive statistical study of age
model errors when using wood, it is simpler and more conservative to utilize a
Gaussian age model error.
Given these potential errors, it is worth considering the modern $^{14}$C age difference
between seawater at the sediment-water interface and the atmosphere. A
measurement of seawater DIC $^{14}$C age close to our core site and depth (at 22°N,
110°W at 598 m), gives a $^{14}$C age of 1240 years BP. Assuming that DIC at this depth
has not yet been seriously impacted by bomb $^{14}$C (Key et al., 2004) this would
predict a pre-bomb wood-to-benthic foraminifera $^{14}$C age difference of 1240 years
BP. This is consistent with our data presented below, where the $^{14}$C age difference
between concurrent wood and benthic foraminifera *P. ariminensis* and *U. peregrina*
varies between this and even larger $^{14}$C age differences (Table 2).
**3.0 Results**
**3.1 Age model and sedimentation rates**
The old coretop age for the LPAZ-21P core (5.3-kyr BP) indicates a poor recovery of
the youngest sediments by the piston core, similar to nearby coring sites on the
Pacific margin (van Geen et al., 2003). The LPAZ-21PG gravity core calendar ages
range from 7954 to 504 years BP, suggesting that it recovered much of the material
missed by the piston core. Both cores give similar sedimentation rates of 16 to 18
cm kyr$^{-1}$ over this Holocene interval (see Figure 3A). The nearby trigger core ET97-
7T gives a slightly lower sedimentation rate for this time interval (pink in Figure 3),
which may result from regional hydrographic differences, different seafloor
dynamics, or sediment recovery based on different coring technology. Core recovery
equipment may also explain differences in downcore sedimentation rates between
the sites (5 cm kyr$^{-1}$ versus 19 cm kyr$^{-1}$ in ET97-7T during the 13-to-15-kyr BP
interval; Figure 3).
A wood $^{14}$C age-constrained age model has only been applied twice before (Broecker,
2004; Zhao and Keigwin, 2018) and it is worth quantifying the suitability of this
approach in our cores. First, we applied a quantitative test: the wood $^{14}$C age must
be older than all coexisting foraminifera $^{14}$C ages. This test included planktic
foraminifera measurements that will be discussed in a following manuscript. The
difference between benthic foraminifera and wood $^{14}$C ages is illustrative of the
effectiveness of this test. The difference between the $^{14}$C age of benthic foraminifera
(*P. ariminensis* and *U. peregrina*) and coexisting, wood that passed our test is
2346±1599 years (n=14) and 2309±1063 years (n=14), respectively (Table 2). Only
comparing wood with foraminifera abundance maxima gives a $^{14}$C age difference of
3353±1957 years (*P. ariminensis*; maximum of 5815 years, minimum of 1077 years,
n=6) and 2697±1117 years (*U. peregrina*; maximum of 4145 years, minimum of
1480 years, n=6). These values are consistent with bottom water at our core sites
that are near or older than the modern, pre-bomb seawater - atmosphere $^{14}$C age
difference of 1240 years (see above). Given these results, we argue that our test for
excluding wood $^{14}$C ages is appropriate, but that unlikely circumstances may have
existed that could hide the timescale of deposition. In the event of a longer-than-
expected time between wood growth and deposition in the sediment, the calendar
age would be biased to younger ages, making benthic foraminifera Δ$^{14}$C values even
more depleted than calculated (Figure 5).
The excellent age model controls provided by wood $^{14}$C provide us with a powerful
(and not always flattering) insight to the sedimentation rates of the Gulf cores. For
example, our wood-constrained calendar ages identify two periods of slow
sedimentation (or possibly hiatus events) in LPAZ-21P (between 22.7- to 19.5-kyr
BP and 12.1- to 9-kyr BP; see grey bars in Figures 3, 4, and 6). The earlier interval is
bracketed by wood-constrained calendar ages while the shallower / more recent
sedimentation rate slowdown begins approximately at the end of the Younger Dryas
or less than ≈12.1-kyr BP.
### 3.2 Foraminifera abundance estimates
The abundance of four benthic and one planktic foraminifera in the LPAZ-21P core
is highly variable with the planktic species *G. bulloides* as high as >6000 g$^{-1}$ of
sediment (Figure 4). The least abundant foraminifera was *P. ariminensis*, which had
peak values just over 200 g$^{-1}$. Abundance of *G. bulloides* and all other planktic
foraminifera (not shown) in these sediments dropped sharply after 12.1-kyr BP—a
loss of planktic foraminifera preservation that is also seen at the nearby California
Undercurrent site (red diamond in Figure 2) (Lindsay et al., 2015). The abundance
of *P. ariminensis* also drops to zero after 12.9-kyr BP, while *U. peregrina* and *T.*
*bradyi* decline to lower, but persistent values ≈2-kyr later. *Bolivina* spp. are known
to persist in low oxygen waters and are the most abundant foraminifera in LPAZ-
21P and LPAZ-21PG sediments for the past 7-kyr.
It is important to identify the abundance of sedimentary foraminifera when
measuring [14]C because the vertical mixing of sediment by macro- and micro-fauna
(bioturbation) can grossly bias the [14]C results (Keigwin and Guilderson, 2009)
causing foraminifera [14]C ages to be older on the shallow side of abundance peaks,
and vice versa. This effect was recently shown for Juan de Fuca Ridge sediments,
where foraminifera [14]C measurements shallower than a large abundance maxima
were biased to "old" [14]C ages (Costa et al., 2018). Below we explore the [14]C age and
Δ[14]C trends for each benthic foraminifera species.
**3.3 Comparing benthic foraminifera [14]C measurements**
Examining the differences in the [14]C age of the four benthic foraminifera species
(Figure 5), we find a maximum 5775 year offset between *U. peregrina* and *T. bradyi*
[14]C age (the former being older). Even though the sample sizes are small (7 to 42),
comparing the preferred epifaunal *P. ariminensis* (Keigwin, 2002) to the other
species suggests that: (1) *Bolivina* spp. [14]C age is older, (2) *T. bradyi* [14]C age is
younger, and that (3) *U. peregrina* gives a [14]C age that is most similar to the
epifaunal species (Table 3; left side).
The comparisons above, however, are likely influenced by bioturbation and a more
appropriate examination would only compare the [14]C ages of foraminifera at
abundance maxima where the influence of bioturbation is minimized (see above).
One drawback to an abundance maximum-only comparison is that it draws from a
smaller pool of observations (e.g., n=2 for the *P. ariminensis* vs. *Bolivina* spp.), which
limits the significance of these statistics. This comparison suggests that—on
average—*U. peregrina* (n=8) and *T. bradyi* (n=4) give similar [14]C ages to epifaunal
species, but with a large (10±861 years) to very large (35±1125 years) range of
variability. On average, *Bolivina* spp. at abundance maxima (n=2) gives an even
older [14]C age difference from the preferred epifaunal species (Table 3; right side).
Despite the monospecies Δ[14]C differences, the glacial-deglacial trends of all four
benthic foraminifera [14]C (corrected for decay and shown as Δ[14]C in Figure 5) from
our cores near the mouth of the Gulf of California (the 'Gulf' sediment core sites) are
depleted relative to the atmosphere during the deglaciation, but are considerably
higher during the Holocene. The shallowest and therefore most recent benthic
foraminifera Δ[14]C are roughly equal to modern DIC Δ[14]C measurements of -173‰ at
the depth of the cores (Key et al., 2004). Error bars denote 1 sigma calendar age and
Δ[14]C errors and symbols represent measurements at abundance maxima. Triangles
with error bars at bottom of each plot indicate the calendar ages and 1 sigma
uncertainties provided by wood dates. The [14]C ages of foraminifera on either side of
abundance peaks can also be "corrected", although this requires an assumption
about bioturbation rates, but this will be the subject of future work.
Each monospecies $\Delta^{14}C$ record in Figure 5A to D is compared with the nearby
benthic foraminifera $\Delta^{14}C$ record from the open Pacific margin of Baja California (a
combination of mixed and mono-species benthic foraminifera on the original age
model; see core locations in Figure 2 and Table 1) (Lindsay et al., 2015; Marchitto et
al., 2007). Additionally, a series of mixed benthic $\Delta^{14}C$ measurements (preliminary
work on ET97-7T where species abundance was not quantified) is shown in Figure
5D. All Gulf benthic foraminifera $\Delta^{14}C$ measurements are compiled in Figure 5E
(black) to illustrate the overall range of values given by the 4 benthic and mixed
species measurements relative to the atmospheric (grey; (Reimer et al., 2013)) and
the Undercurrent site $\Delta^{14}C$ (red). As can be seen by these multiple views of the
dataset, all benthic foraminifera $\Delta^{14}C$ trends at both Gulf and Undercurrent sites
shift to lower values after 20-kyr BP. These depletions relative to atmospheric $\Delta^{14}C$
are very large, but even lower $\Delta^{14}C$ values are observed for intermediate depth
sediment core sites in the eastern equatorial Pacific (Stott et al., 2009).
**3.4 Influence of macrofaunal consumption and excretion on sediment $^{14}C$**
**ages?**
In a single interval from 106 to 110 cm of the LPAZ-21P core, which was predicted
to be ≈25.5-kyr based on interpolation from our Bayesian statistical age model, the
wood and benthic and planktic foraminifera $^{14}C$ ages were conspicuously younger
than expected, giving a $\Delta^{14}C$ value well above the contemporary atmosphere
(956.3‰; see circle in Figure 6). In fact, wood found within this sedimentary
interval suggests a calendar age of only 19.2-kyr BP, giving a much lower *U.*
*peregrina* $\Delta^{14}C$ of -88.6‰. This lower $\Delta^{14}C$ value is consistent with values for the
wood-constrained calendar age (see square in Figure 6). If these anomalous but self-
consistent observations are not simply a result of human error (mislabeling or other
sampling problem) they may indicate the presence in this interval of "zoophycos" or
the remnants of downward-burrowing macrofauna (as was suggested by (Lougheed
et al., 2017)). By consuming and later excreting sedimentary material, these worms
are able to move 'younger' sedimentary components deeper in the sediment column,
though if this is the cause, the self-consistency of our $^{14}C$ measurements in this
reworked interval (where microfossil and wood $^{14}C$ ages suggest an undisturbed
sample) is surprising.
**3.5 The stable isotopic composition of oxygen ($\delta^{18}O$) and carbon ($\delta^{13}C$)**
The epifaunal benthic foraminifera (*P. ariminensis*) $\delta^{18}O$ and $\delta^{13}C$ measurements in
Figure 7 uses new and published data from LPAZ-21P (Herguera et al., 2010), but on
our wood-constrained age model. As previously reported, intermediate depth $\delta^{13}C$
shows little variability between the Last Glacial Maximum (LGM) and Holocene at
the depth of this core (624 m; (Herguera et al., 2010)). Benthic foraminifera $\delta^{18}O$ has
similar magnitude of change to benthic $\delta^{18}O$ for the nearby Undercurrent core sites
(Figure 7 in (Lindsay et al., 2016)), although the Undercurrent benthic $\delta^{18}O$ increase
to Holocene values may lag the Gulf site values (compare (Lindsay et al., 2016) with
Figure 7).
**4.0 Discussion**
Based on the work presented here, the trend towards an extreme lowering of
intermediate-depth benthic foraminifera $\Delta^{14}$C in the subtropical northeastern
Pacific (the California Undercurrent site in Figures 1, 2, and 5) (Lindsay et al., 2015;
Marchitto et al., 2007) cannot be explained by species biases, bioturbation, or poor
age model controls (Figure 5). This statement is supported by our $^{14}$C
measurements of the epifaunal benthic foraminifera *P. ariminensis*—a species
known to provide the best record of seawater carbon at the sediment-water
interface (Keigwin, 2002)—and several commonly used infaunal benthic
foraminifera from sediment cores "upstream" of the canonical record of these
extreme $\Delta^{14}$C observations (Figures 1 and 5). Our measurements indicate that even
though the potential variability between infaunal and the preferred epifaunal
species' $^{14}$C ages is relatively large (several hundred years; Table 3), the average $^{14}$C
age difference at foraminifera abundance maxima is <100 years, and the overall
trend towards extremely low $\Delta^{14}$C during the deglaciation cannot be explained by
bioturbation and persists regardless of species.

**4.1 Comparing Gulf and Undercurrent site deglacial records**
Our Gulf sediment core observations indicate that the mixed-species $\Delta^{14}$C
measurements from the Undercurrent sites shown in Figures 1 and 5 are largely
accurate, although the higher values that form the middle of this 'W' shaped
anomaly (from ≈15- to 13-kyr BP) are not obviously reproduced by any of the 4
mono-species benthic foraminifera $\Delta^{14}$C. It is possible that this and other some
smaller-scale features of a mixed benthic $\Delta^{14}$C record reflect the bias of a particular
species and/or the influence of bioturbation in our lower sedimentation rate sites.
For example, the benthic foraminifera *T. bradyi* is a possible suspect for biasing
mixed benthic $\Delta^{14}$C measurements because it is relatively large, dense, and
sometimes has large deviations to younger $^{14}$C ages than the other species (Figure
5). Nevertheless, the overall agreement between the independently derived
Undercurrent and Gulf records lend credence to the methods used to construct the
age model by Marchitto et al., (2007) and tested by Lindsay et al., (2016). We should
note that we cannot explain the large offset between the records from 30-to-25-kyr
BP, although this comparison only includes one observation from the Undercurrent
sites.

The similar $\Delta^{14}$C trends at both Undercurrent and Gulf sites despite sedimentation
rate differences and sediment core hiatus lends additional support for the
robustness of the $\Delta^{14}$C trends (Lindsay et al., 2016) and against events such as the
large-scale—and far-fetched—redeposition of sand-sized sedimentary components.
In principle, the circulation of bottom waters from the Gulf to the Undercurrent
sediment core sites could allow for redeposition of benthic foraminifera with much
older $^{14}$C ages, but a much larger reworked component (and hence much older
benthic foraminifera [14]C ages) would logically be expected at the "upstream" Gulf
sites. In fact, sedimentary redeposition should be amplified at the lower
sedimentation rate Gulf site, but significantly lower benthic foraminifera $\Delta^{14}C$ is not
observed for any of the species at the Gulf sites.
These findings allow us to now focus our questions on two potential explanations
for the extreme depletions of benthic foraminifera $\Delta^{14}C$ observed during the
deglaciation: (1) it is a diagenetic signal imparted onto both epifaunal and infaunal
foraminifera after burial or (2) it reflects a real change in seawater $\Delta^{14}C$ during the
deglaciation.

**4.2 Can diagenesis explain the low deglacial $\Delta^{14}C$?**
Investigating the potential for diagenetic alteration of benthic foraminifera $\Delta^{14}C$, we
are not concerned about the newly observed coupling between carbonate
dissolution and precipitation (Subhas et al., 2017), which only involves a few
monolayers of surface carbonate. Instead, producing the extreme $\Delta^{14}C$ lowering
observed at Undercurrent and Gulf sites (Figure 5) and other sites around the globe
(Bryan et al., 2010; Stott et al., 2009; Thornalley et al., 2011) requires the
precipitation of depleted [14]C on or within the foraminifera test is required.
This authigenic calcium carbonate formation and foraminifera [14]C content has been
examined in several ways. For example, benthic foraminifera from the eastern
equatorial Pacific give some of the lowest observed deglacial $\Delta^{14}C$ values (-609‰),
but Scanning Electron Microscope images show no authigenic carbonate on benthic
or planktic foraminifera (Stott et al., 2009). Calcium carbonate overgrowth (via the
conversion of $CaCO_3$ to $CaSO_4$ (gypsum)) was observed in Santa Barbara Basin
sediments (Magana et al., 2010), but would not influence the [14]C content of the
microfossil. What's more, extreme [14]C depletions of mixed benthic foraminifera
from this and other sites were found to be biased by *Pyrgo* spp., which are
inexplicably depleted in [14]C (Ezat et al., 2017). Other work suggests *younger-than-*
*expected* [14]C ages from the precipitation of carbonate onto foraminifera tests after
core recovery (Skinner et al., 2010). Cook et al., (2011) observed anomalously low
foraminifera $\Delta^{14}C$, high $\delta^{18}O$, and low $\delta^{13}C$ was consistent with authigenic carbonate
precipitation from methane. Wycech et al., (2016) also compared the [14]C ages of
translucent and opaque mono-specific planktic foraminifera from the same
sediment horizons and found the opaque foraminifera (thought to contain
authigenic carbonate) had [14]C ages more than 10,000 years older than the
translucent tests.
Neither the Gulf nor the Undercurrent site benthic foraminifera measurements
display the telltale signs of simultaneous $\Delta^{14}C$, $\delta^{18}O$, and $\delta^{13}C$ anomalies seen by
Cook et al., (2011) (see Figure 7). What's more, the planktic $\Delta^{14}C$ values from the
Undercurrent site do not show anomalous depletion during the deglaciation
(Lindsay et al., 2015), which is expected for post-depositional alteration / authigenic
carbonate formation. It is possible that a completely different process of authigenic
carbonate formation is occurring in the subtropical eastern Pacific, but we cannot
elaborate on what this mechanism might be. It is possible that authigenic carbonates
are removed from the foraminiferal test during the 10% acid leaching pre-treatment
at KCCAMS (see Methods), although selected pre-treatment tests did not
significantly alter the $^{14}$C ages. This pretreatment was not used in the Wycech et al.,
(2016) comparisons, but will be examined in our future studies.
Finally, given the near identical deglacial $\Delta^{14}$C trends at the Undercurrent and Gulf
sites despite very different sedimentation rates (20-30 cm kyr$^{-1}$ at the Undercurrent
versus 1-to-5 cm kyr$^{-1}$ at the Gulf; Figure 3) it would be surprising if the same
depleted $\Delta^{14}$C trends were of diagenetic origin. This is because a faster
sedimentation rate will decrease the potential for authigenic mineralization by
decreasing the exposure time of the foraminifera. This reduction in exposure time
would apply to both the microfossil's exposure at the sediment-water interface and
at sediment depths favorable to authigenic carbonate precipitation. Thus, while the
potential influence of authigenic carbonate on the primary foraminifera record is an
important area of research that deserves further study, the similarity of the
Undercurrent and Gulf records argues against contamination from authigenic
carbonate precipitation as the major influence on these benthic foraminifera $\Delta^{14}$C
values.
**5.0 Conclusions**
If the extreme deglacial depletion of benthic foraminifera $\Delta^{14}$C at these northeastern
Pacific sites cannot be explained by species or habitat bias, bioturbation, or poor age
model control, the remaining explanation is that they reflect a change in seawater
DIC $\Delta^{14}$C. Looking to other proxy systems, deep-sea coral $\Delta^{14}$C in the North Atlantic
and Southern Ocean—archives with excellent age model control and different
diagenetic influences—also display depleted deglacial $\Delta^{14}$C during the deglaciation
(Adkins et al., 1998; Burke and Robinson, 2012; Chen et al., 2015; Robinson et al.,
2005). However, the deep-sea coral $\Delta^{14}$C depletion have a different timing and are
not as extreme as observed for the Gulf of California and California Undercurrent
sites (Figure 5E).
A leading candidate among the potential explanations for these and other
intermediate depth records (Bryan et al., 2010) is the deep-sea sequestration and
flushing of carbon through the intermediate depth ocean (Basak et al., 2010; Du et
al., 2018; Lindsay et al., 2016; Marchitto et al., 2007). This interpretation is plausibly
consistent with $^{14}$C records from only a few sites, such as the deep Southern Ocean
(Barker et al., 2010; Skinner et al., 2010) and deep Nordic Seas (Thornalley et al.,
2015). However, using an 18-box geochemical ocean-atmosphere model to simulate
glacial-interglacial ocean circulation and carbon cycling, Hain et al., (2011) argue
that matching the observed $\Delta^{14}$C depletions in the intermediate depth, Northern
Hemisphere sites requires unrealistic changes in ocean chemistry (e.g., lower
surface ocean alkalinity) and ocean dynamics (i.e., mixing). Specifically, to
appropriately "age" deep-sea $^{14}$C requires deep-sea anoxia, which is not observed.
Furthermore, the release of this deep-sea $^{14}$C to intermediate depths would
dissipate much quicker than the several thousand year anomaly shown in Figure 5E.
An alternative explanation involves the addition of $^{14}$C-depleted carbon via mid-
ocean ridge (MOR) volcanism (Ronge et al., 2016), which is indirectly supported by
evidence for increased MOR activity (Lund, 2013; Middleton et al., 2016; Tolstoy,
2015). The locations and depths of the extreme benthic foraminifera $\Delta^{14}$C lowering
are also suggestive of a MOR influence, given their proximity to the East Pacific Rise
/ Gulf of California (Marchitto et al., 2007; Ronge et al., 2016; Stott et al., 2009; this
study), the Red Sea (Bryan et al., 2010), and Mid-Atlantic Ridge (Thornalley et al.,
2011). However, this hypothesis of enhanced carbon flux from seafloor volcanism
must also explain the many intermediate-depth sites that do not show anomalous
deglacial $\Delta^{14}$C depletions (Broecker & Clark, 2010; Cléroux et al., 2011; De Pol-Holz
et al., 2010). Furthermore, this proposed carbon addition must have been associated
with an alkalinity addition, without which the increased seawater $CO_2$
concentrations and therefore lower seawater pH would have caused a global-scale
carbonate dissolution event (Lindsay et al., 2016; Stott and Timmermann, 2011).
In summary, our work strongly suggests that at least for the Gulf of California and
adjacent Pacific sites, the foraminifera $\Delta^{14}$C proxy records real $^{14}$C changes in
deglacial intermediate depth seawater DIC, but the question of what caused those
changes remains open. Careful examination to confirm or disprove the fidelity of the
benthic foraminifera $\Delta^{14}$C on a case by case basis will be a critical part of building a
reliable body of data to identify the controls on glacial-interglacial marine carbon
cycling.

**Acknowledgments:** C. Bertrand, A. Hangsterfer (SIO Core Repository), H. Martinez,
N. Shammas, M. Ayad, M. Rudresh, A. De la Rosa, J. Troncoso, J. DeLine, J. Sanchez, C.
Manlapid, M. Chan, as well as T. Marchitto and two anonymous reviewers.

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

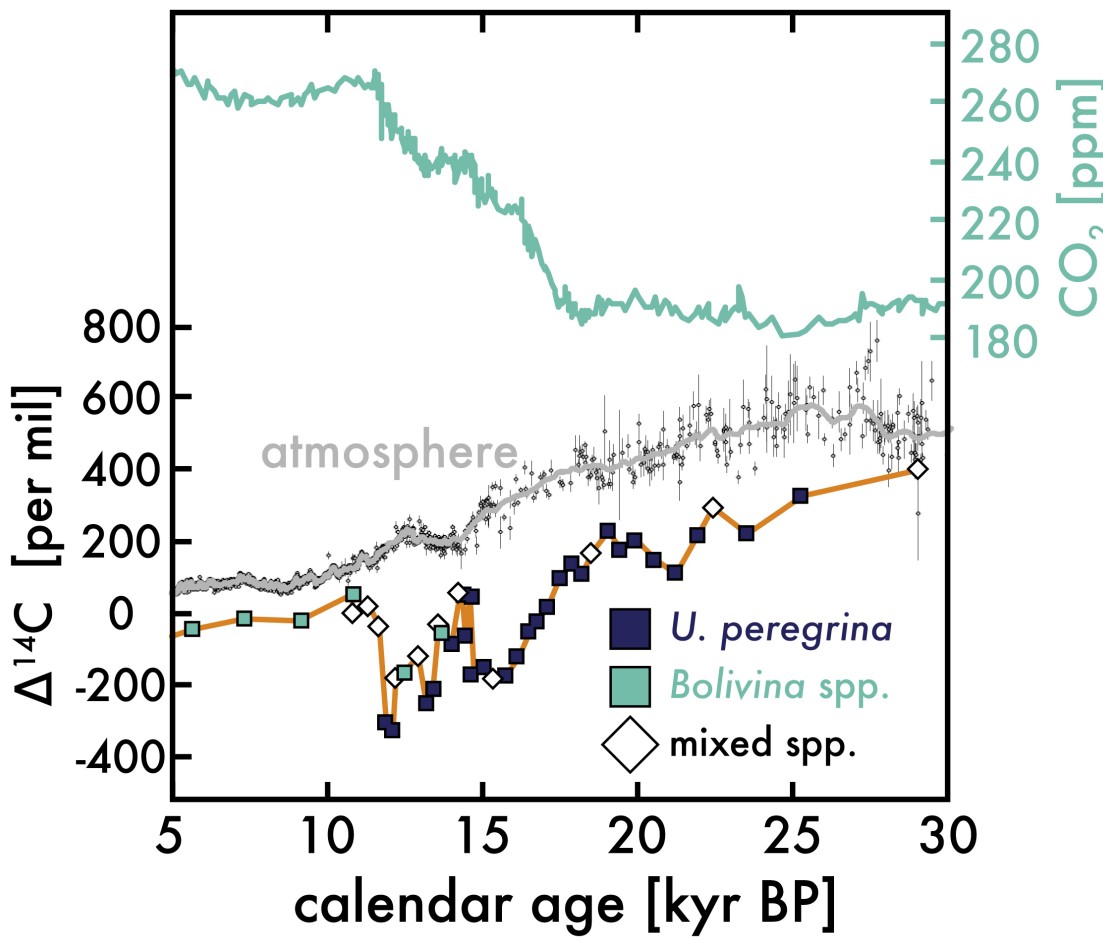

**Figure 1.** Atmospheric carbon dioxide ($CO_2$) concentrations (top; blue) (Ahn and
Brook, 2014; MacFarling Meure et al., 2006; Marcott and Shakun, 2017; Monnin,
2001), atmospheric $\Delta^{14}C$ (middle: blank symbols observations; gray line is
smoothed average) (Reimer et al., 2013), and benthic foraminifera $\Delta^{14}C$ from
sediment bathed in California Undercurrent water (orange; see Table 1 and maps in
Figure 2) (Lindsay et al., 2015; Marchitto et al., 2007). These observations are
shown from 30-to-5-kyr BP (BP = before 1950).


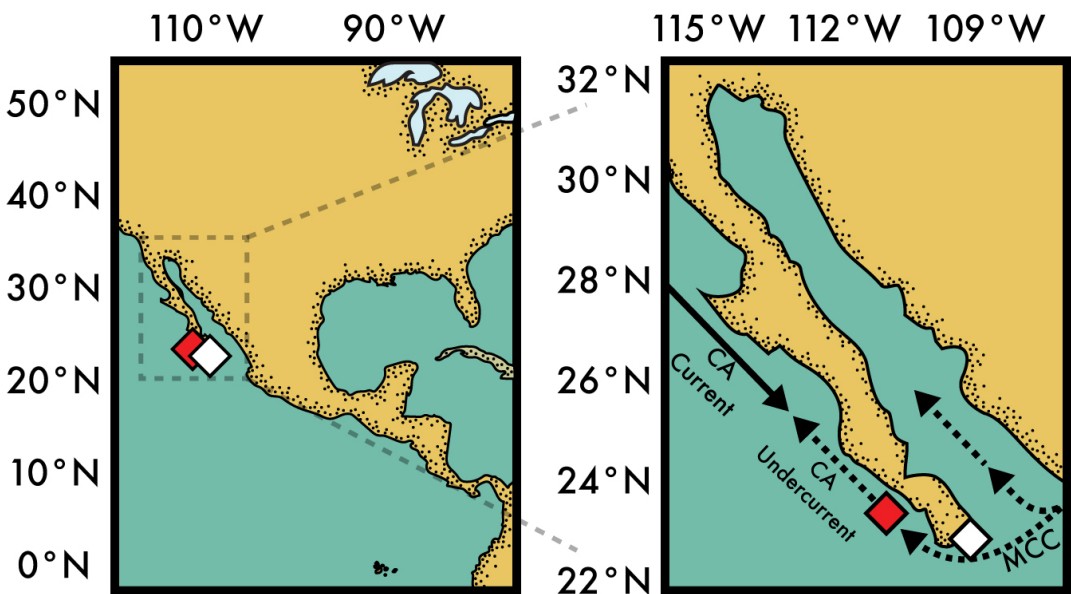


**Figure 2.** Maps of sediment core sites (diamonds; see Table 1) and ocean circulation
(arrows: solid = surface; dashed = near seafloor). The foraminifera radiocarbon
measurements in Figure 1 are from core sites at the red diamond (Marchitto et al.,
2007; Lindsay et al., 2016). See Table 1 for details on site locations. Note that the
subsurface Mexican Coastal Current (MCC) flows between 200 to ≈700 m and feeds
subsurface water into both the Gulf of California and California Undercurrent
(Gómez-Valdivia et al., 2015)—waters that bathe both core sites.


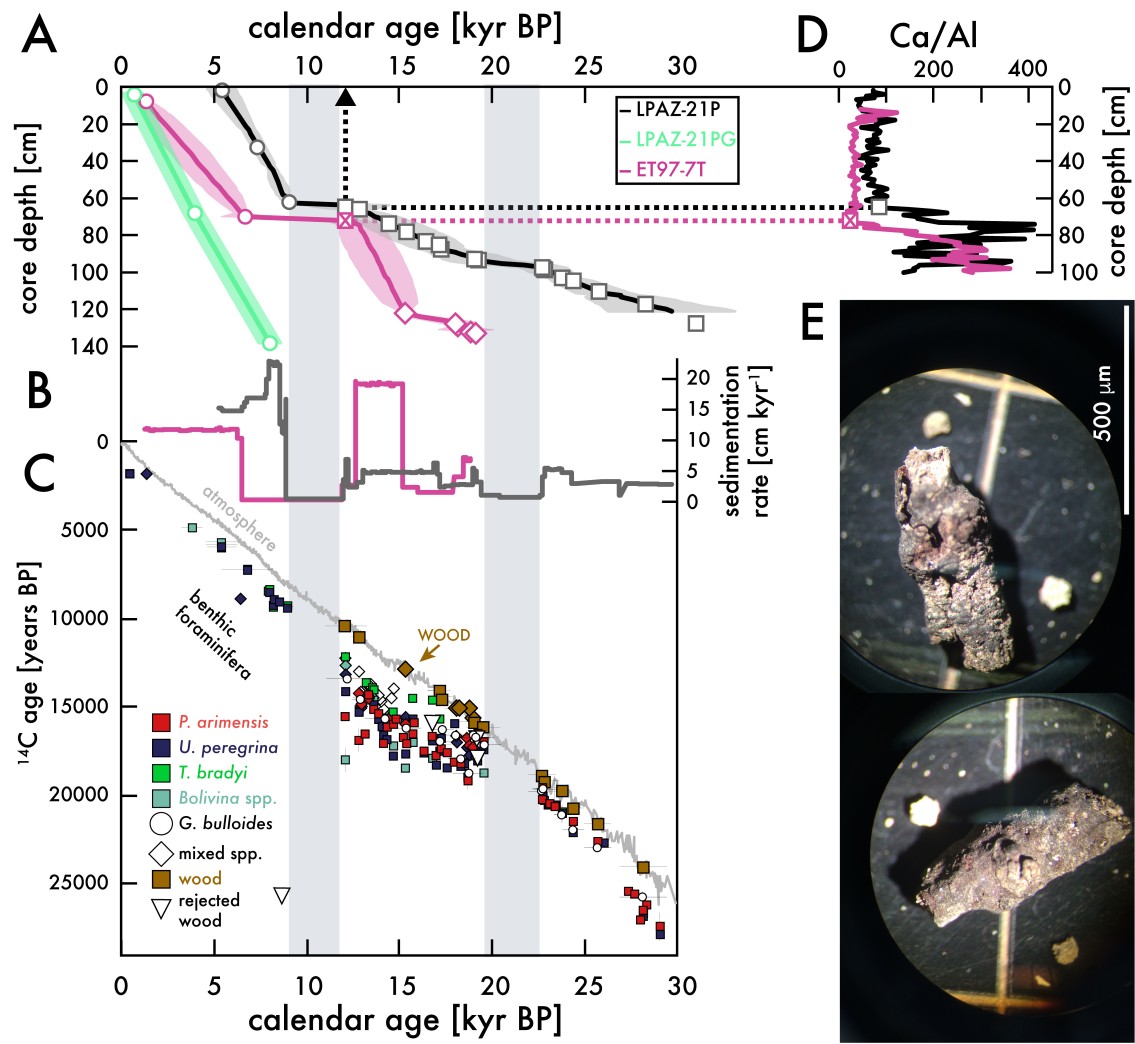


**Figure 3.** (A) Sediment core depth versus calendar age. Age model constraints are based on wood $^{14}$C (squares and diamonds), stratigraphic correlation ("X"; see (D)), and U. peregrina $^{14}$C corrected for reservoir age (circles). (B) Sedimentation rate versus calendar age. (C) The $^{14}$C age of atmospheric $CO_2$, foraminifera, wood, and rejected wood (see legend) versus calendar age for ET97-7T (diamonds) and LPAZ-21P/ LPAZ-21PG (squares). See Figure 1 and Table 1 for locations. (D) Ca/Al for ET97-7T (pink) and LPAZ-21P (black) was measured using X-Ray Fluorescence (see Methods). Lower Ca/Al for the uppermost sediment (beginning at the arrows) is coincident with loss of calcium carbonate microfossils and an overall darkening of sediments at these and other sites in the region (van Geen et al., 2003). We use this stratigraphic feature to tie the age model for both sites (dashed arrow between "X"s). (E) Examples of wood found within sediment core LPAZ-21P (see scale).


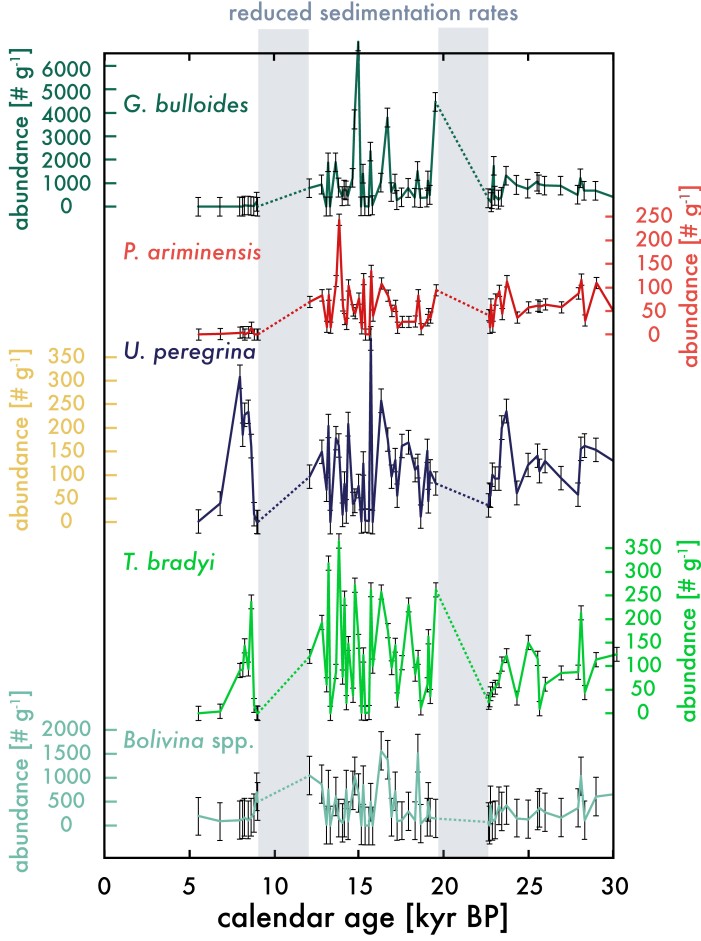


**Figure 4.** Foraminifera abundance at site LPAZ-21P for planktic (*G. bulloides*) and

benthic species (all others). Error bars represent 2 times the typical standard

deviation for replicate counts.



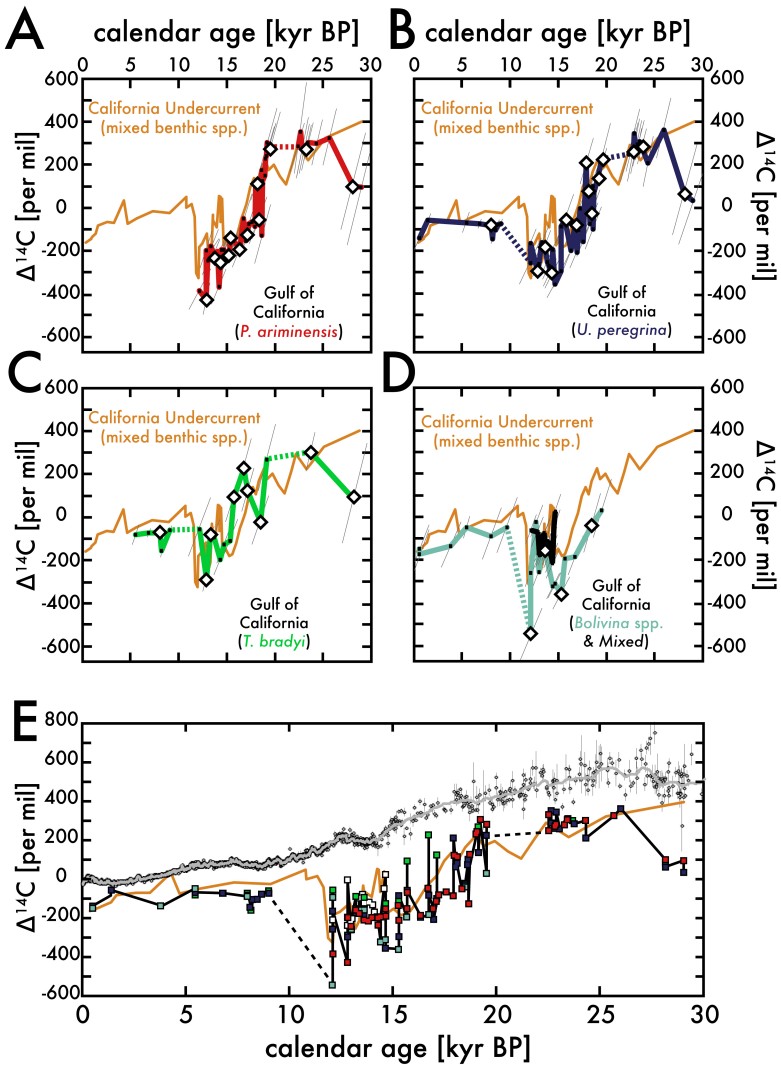


**Figure 5.** The $^{14}$C isotopic composition (corrected for decay as $\Delta^{14}$C) for Gulf of California benthic foraminifera mono-species (A-D) and mixed species (D) are compared with mono- and mixed-species benthic foraminifera $\Delta^{14}$C measurements from the California Undercurrent sediment core site (orange) (Lindsay et al., 2015; Marchitto et al., 2007) over the past 35,000 years BP. See core locations in Figure 2. Canted error bars take into account measurement and age model errors (Stuiver et al., 2017). Diamonds indicate $\Delta^{14}$C at foraminifera abundance maximum. The $\Delta^{14}$C of all Gulf benthic foraminifera (see Figure 3 legend) is shown in (E) alongside atmospheric $\Delta^{14}$C (grey) (Reimer et al., 2013) and California Undercurrent site $\Delta^{14}$C (orange). Modern $\Delta^{14}$C near the depth of the Gulf and Pacific sites is about -173‰ (Key et al., 2004).

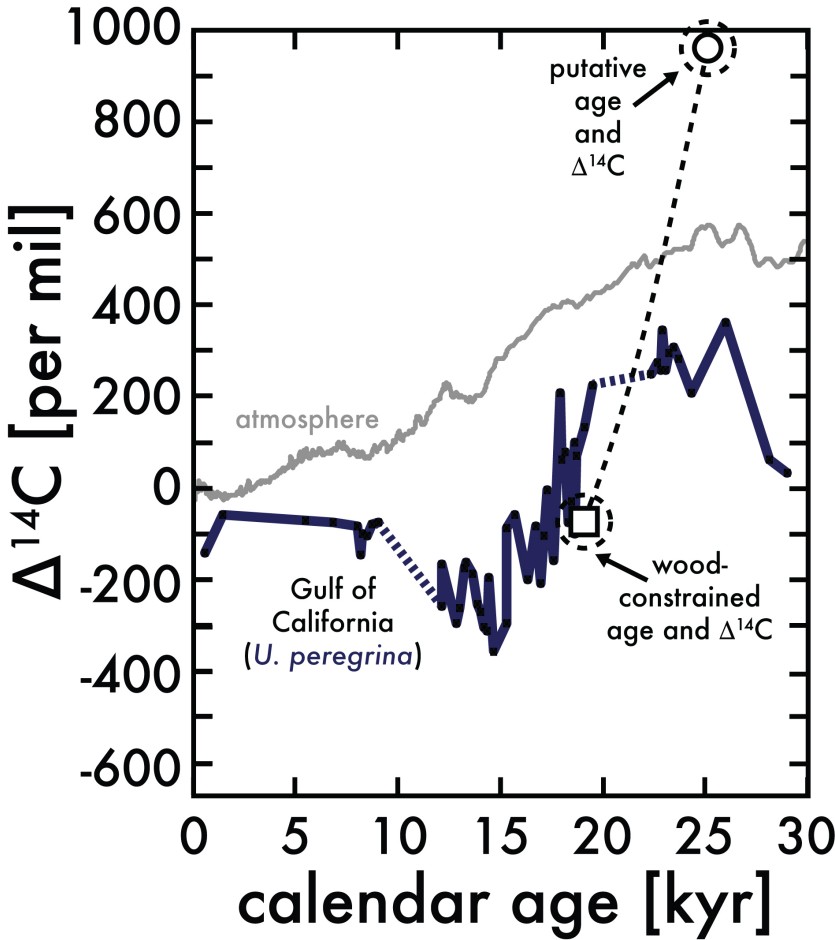

871

**Figure 6**: Comparing Gulf of California *U. peregrina* Δ14C with anomalous values

based on the Bayesian calendar age (circle) based on sediment depth and the wood-

constrained calendar age (square). If this anomaly was not the result of human error

(mislabeling of the sample's depth), then this may suggest the influence of

macrofauna. See text for more details.

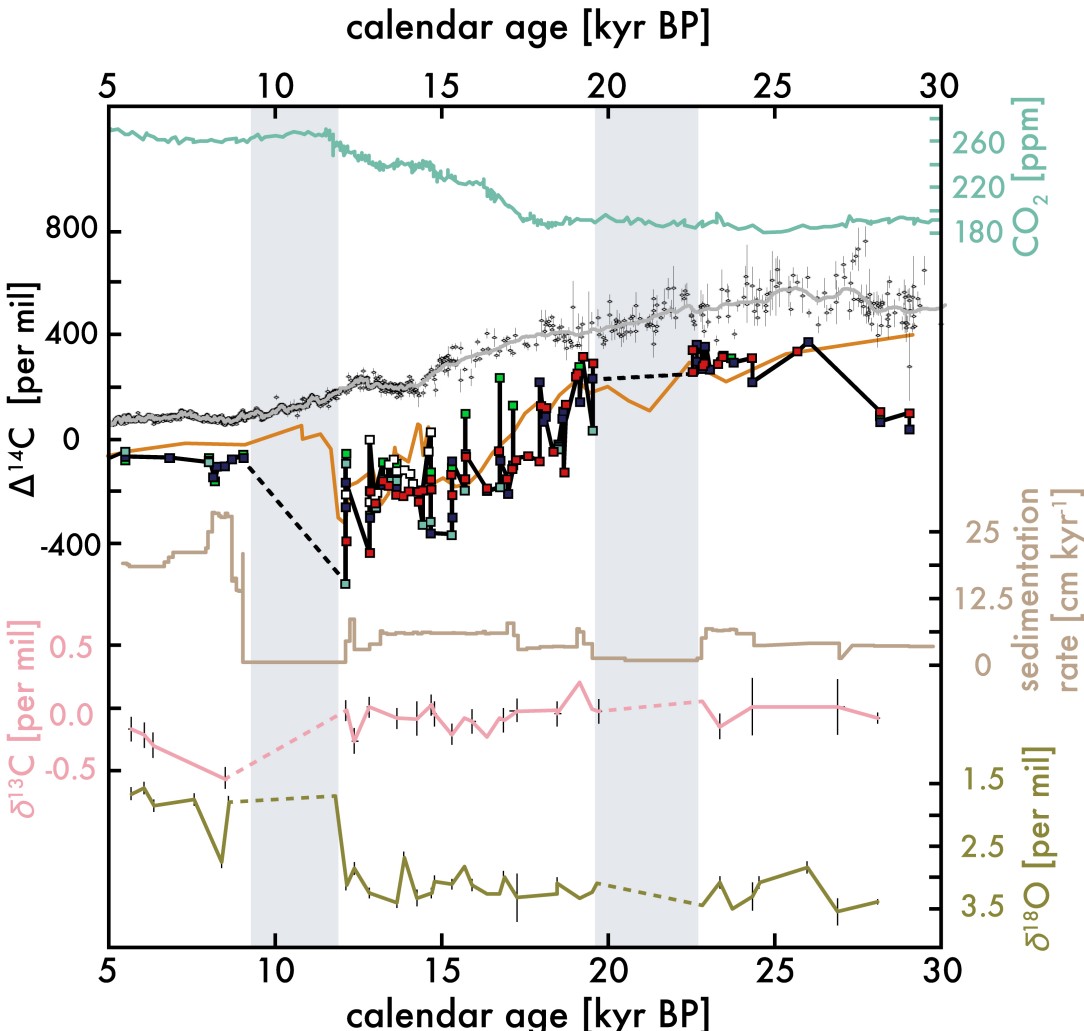

**Figure 7.** From top to bottom: atmospheric carbon dioxide ($CO_2$) (blue; same as
Figure 1), atmospheric $CO_2$ $\Delta^{14}C$ (grey; same as Figure 1), mixed and mono-species
benthic foraminifera $\Delta^{14}C$ from the California Undercurrent site (red) (Lindsay et al.,
2015; Marchitto et al., 2007), all mono-species benthic foraminifera $\Delta^{14}C$ from near
the mouth of the Gulf of California (black; this study), sedimentation rate of LPAZ-
21P (see Figure 3), benthic foraminifera *P. ariminensis* $\delta^{13}C$ (pink) and $\delta^{18}O$ (green)
from this study and Herguera et al., (2010).

**Table 1.** Latitude, longitude, depth below modern sea surface, and modern
dissolved inorganic carbon (DIC) $\Delta^{14}C$ (26) at the sediment-seawater interface for
sediment cores discussed in this study.

| Table 1 | latitude [°N] | longitude [°W] | depth [m] | modern DIC $\Delta^{14}C$ [‰] at this depth |
|---|---|---|---|---|
| LPAZ-21P / LPAZ-21PG | 22.9 | 109.5 | 624 | -148 |
| ET97-7T | 22.9 | 109.5 | 640 | -148 |
| MV99-MC19/GC31/PC08 | 23.5 | 111.6 | 705 | -148 |


**Table 2.** The difference between benthic foraminifera and concurrent wood $^{14}C$
ages for all measurements ("ALL") and only at abundance maxima (see Figure 4).

| | The difference between *P. ariminensis* - wood [$^{14}C$ years] | | The difference between *U. peregrina* - wood [$^{14}C$ years] | |
|---|---|---|---|---|
| | ALL | abundance maxima | ALL | abundance maxima |
| AVERAGE | 2346 | 2488 | 2309 | 2697 |
| STDEV | 1599 | 1791 | 1063 | 1117 |
| n | 14 | 11 | 13 | 6 |


**Table 3.** Comparison between benthic foraminifera $^{14}C$ ages for all measurements
("ALL") and only at abundance maxima (see Figure 4).

| | The difference between *P. ariminensis* - *U. peregrina* [$^{14}C$ years] | | The difference between *P. ariminensis* - *T. bradyi* [$^{14}C$ years] | | The difference between *P. ariminensis* - *Bolivina* spp. [$^{14}C$ years] | |
|---|---|---|---|---|---|---|
| | ALL | abundance maxima | ALL | abundance maxima | ALL | abundance maxima |
| AVERAGE | -104 | 10 | 826 | 35 | -857 | -1407 |
| STDEV | 759 | 861 | 1484 | 1125 | 939 | 597 |
| n | 42 | 8 | 7 | 4 | 11 | 2 |
