# Peer review of "Extreme lowering of deglacial seawater radiocarbon recorded by both epifaunal and infaunal benthic foraminifera in a wood-dated sediment core"

_Climate of the Past, 2018_

## Referee Comment (RC1) · T. Marchitto (Referee) · 16 Aug 2018

Rafter et al. present a new record of deglacial intermediate water radiocarbon from the southern tip of Baja California. This provides a direct test of the fidelity of the nearby (California Undercurrent, CU) Marchitto et al. record, which was originally proposed to track the deglacial release of aged carbon from the deep ocean, hypothetically via the Southern Ocean and AAIW. The CU result has been questioned because of the lack of a similar signal in other 'expected' locations along the AAIW flow path, and because of the inability of a box model to simulate such a strong signal in the face of mixing with better equilibrated waters. It has been suggested that the CU record may suffer from

artifacts, or that it may record a local release of geologic carbon.

The present record has two principal strengths: it has wood-based 14C age control, and it compares different species of benthic foraminifera, including an epifaunal taxon. The general agreement with the CU record therefore provides a strong argument against some of the hypothetical artifacts that have been previously invoked. I think this is an important result, and I support publication after substantial revisions. (Parenthetical numbers below refer to page/line numbers.)

First, I suggest that the authors place greater emphasis on the novelty and robustness of their wood-based age model. Only very recently has another such wood-based record been published, claiming to be "the first high-resolution record that is free of the dating uncertainties common in marine sediment records" (Zhao and Keigwin, 2018, Nature Communications). The greater emphasis could begin with the title, by appending something like "in a wood-dated core" to the end of it. The value of wood can also be emphasized in the Introduction, where the uncertainties of foram radiocarbon work are enumerated (beginning at 2/33). The first and foremost paragraph there should be about calendar age control, including uncertainties in planktic foram habitat and reservoir age, and (in the case of the CU record) assumptions about temporal correlations to other records. Given the novelty of the wood age model, I would also like to see a more explicit presentation of the rejected dates, including a figure. I do not have an intuition for how wood should behave in a marine core. Zhao and Keigwin suggest that wood will only float for months, and is hence quickly buried; but their core had a much higher sedimentation rate than the present study, so I wonder if the greater number of rejections here might be related to residence time at the seafloor before burial, or its lack of proximity to a river mouth? The accepted dates in Fig. 3 look very nice stratigraphically, but I'd like to see what the five rejected dates look like. The principal detail given is that they were older than coexisting foraminifera. The statement about "macrofauna consumption" (6/5) is nebulous, and it is not cleared up later (9/1) without the aid of a figure.

A key assigned date in core ET97-7T is based on a color change that was wood-dated in the very nearby core LPAZ-21P. This assignment produces a somewhat alarming increase in the sedimentation rate of ET97-7T, in comparison to LPAZ-21P. As such, it needs to be better defended. The color change is described as abrupt in LPAZ-21P but no description is given for it in ET97-7T. It appears to coincide with a hiatus (or sharp drop in sed rate), so nailing the true age of the transition in the core may be difficult. Based on my experience with other Baja cores, I suspect the color change occurs at the start of the Holocene (∼11.7 ka, admittedly subject to assumptions about temporal correlation with Greenland), but mud of that exact age may well be missing. I guess that the 12.1 ka date in LPAZ-21P sits in the lighter colored mud, meaning the depth assigned to that age in ET97-7T (and hence the mud below it) is also light? In that case, that depth is likely no younger than 11.7, a hypothetically small uncertainty that would not make the high sed rates go away. In any case, reflectance records or core photos would help assure me that this age assignment is valid. Because this interval of ET97-7T is not constrained by wood dates, I suggest that the benthic dates in the lower panel of Fig. 3A be color-coded by core, so that one can see where the ET97-7T analyses sit.

The authors devoted considerable effort to counting foram abundances "to account for bioturbation" (3/23) but I do not see where they actually did very much with the abundance data. They compare inter-species ages overall and on abundance maxima (8/6), but n for the latter is unfortunately small. D14C values from abundance maxima are plotted differently in Fig. 5; can the authors say anything about whether off-maxima values are consistent with bioturbation? Since bioturbation is one of the chief potential bogeymen here, I think it deserves its own section in the Discussion, namely "Can bioturbation explain the low deglacial D14C?" A key point is that the wood dates do not support a wholesale bioturbational artifact (and likewise the planktic dates in Lindsay et al. 2015). Perhaps bioturbation can explain some of the scatter in the benthic dates but not the overall deglacial pattern? It is worth noting explicitly that the noise is likely affected by the low sed rates in comparison to the CU site. Given this noise, I'm not

convinced that the lack of higher D14C values forming the middle of the "W" (10/3) is significant.

The interspecies age offsets are interesting, and sometimes alarming. The CU dates are described as "mostly mixed benthic species" (8/20) but that's misleading: we had 29 Uvigerina, 10 Bolivina, and 21 mixed (please clarify in text and Fig. 5). The new cores' interspecies offsets are discussed in terms of 14C age, but only shown as D14C. I suggest that the first data figure should show the 14C ages for each core, by species, versus depth in core (probably it could be combined with the wood date figure that I request). That would be the clearest demonstration of the interspecies offsets, and the potential for bioturbational mixing to explain some of them. It might also be nice to plot the D14C in Fig. 5E as species-color-coded symbols rather than a single black line. The next question is, if not caused by bioturbation, could interspecies differences be real? Are the quoted average age differences (8/4 and Table 2) actually significant? Pore waters are raised in the Intro (3/5), but what direct (non-foram) evidence is there that shallow pore waters can be significantly different age than bottom waters (citations?). Is it plausible that diagenesis (section 4.2) could affect taxa differently due to, e.g., surface texture? On the topic of diagenesis, it seems to me that another argument against that being a dominant effect is that if diagenetic old carbon were somehow migrating from deeper pore waters, it would likely have a pretty different age between the CU site and the present site, given the very different sed rates (cf. 11/24).

The manuscript mostly punts on the question of where the old carbon is coming from (Conclusions), and I don't fault them for that because it remains a puzzling problem. I think diagenesis should not be raised again at the start of this section (12/3) because it was just addressed in the previous section. I think the 'similarity' of distant sites (Southern Ocean, North Atlantic) is overstated; really only the Arabian Sea (Bryan et al., 2010) looks very similar. The lack of any deglacial signal at some key sites, like Chile (De Pol-Holz) and now Colombia (Zhao and Keigwin), could be reiterated. Consider citing another paper just out, Du et al. (2018) Nature Geoscience, which

discusses a possible flushing of the deep Pacific during deglaciation. Rafter et al. are correct to point out that geologic carbon would need to be buffered. This is true not only if the geologic release were "global-scale" (12/21) but even if it were localized to places like Baja California: the DIC addition to local seawater would need to be huge, causing dissolution if not buffered (see Lindsay et al. 2016, p. 1113, for a rough calculation). Assuming buffering is possible, could the lack of a signal at some intermediate sites (12/18) just be because they are not near ridges/vents?

There is a slight muddling of DD14C and age in this manuscript. Part of the CU DD14C discussed in the intro is simply due to higher atmospheric D14C and not water mass aging (cf. 2/14) (see Lindsay papers for magnitude). The present wood-benthic age difference could be caused by intermediate water aging but not by higher atmospheric D14C (6/22), which affects un-normalized DD14C but not age differences.

Additional comments by line:

2/9: The CU core has only one DD14Cvalue >500 per mil; say >400 instead.

2/20: Cite Broecker and Barker somewhere, but not here: they did not mechanistically link the deglacial 14C drop to the CO2 rise, but rather were not convinced of the sign of the abyssal reservoir's impact on pCO2.

2/24: "Equal to" is unlikely due to mixing ala Hain. Stick with "lower than" and say why.

2/25: "Lower" than deglacial intermediate waters, or lower than today at those locations?

2/29: Clarify that these are places where the signal would be expected if it was carried by AAIW (the signal would not be expected in ALL intermediate depth waters).

2/30: Is this Hain critique different from the mixing argument two sentences before? If so, briefly explain.

3/10: Note that this is largely attributed to the mysterious Pyrgo problem.

3/13: I'm not sure "infrequently taken into consideration" is fair. Everyone "considers" biuoturbation, but we infrequently quantify it?

3/19: I don't really see this paper as a "wide-ranging test of the fidelity of the benthic foraminifera D14C proxy." It is more directly a test of the Marchitto result. I think the fidelity of the proxy will remain pretty core-dependent.

4/6: What size fraction was counted?

4/27: Are these numbers mass of carbon or mass of CaCO3/wood?

8/15: Holocene D14C is noted to be similar to modern, but isn't that forced to be the case (at least for Uvigerina, which dominates the observations) due to using Uvigerina for the Holocene age model?

9/16: "Around 13-kyr" is not precise enough for the CU d18O drop. There are step changes at the start of the Bolling (14.6, halfway to Holocene values) and end of YD (11.7). Caveat: those dates are age model dependent. But it looks like you have the end-YD jump, with a muted/smoothed Bolling earlier.

9/29: This statement about interspecies age offsets being modest does not match the previous discussion nor the Table. Am I misunderstanding?

10/14: Not sure why van Geen is cited here?

10/15: It sounds like you're talking about transporting sand-sized benthic forams from one site to the other (at the same water depth, not downslope), which is far-fetched. Clarify what you mean here.

11/6: Say why you think Pyrgo is so old (or say that it's not known why).

12/11: Clarify that the unrealistic bit about alkalinity (in Hain's model) is that alkalinity gets trapped in the glacial (not deglacial) deep ocean at the expense of the surface, hence raising pCO2.

Figure captions: Please provide descriptions for the different symbols, so the reader does not have to search through the text to interpret figures.

---

## Author Comment (AC1) · 21 Sep 2018

We thank Dr. Marchitto for this thoughtful and useful review. We have adjusted the manuscript in response to every comment, either by modifying text and figures or adding a new figure (as suggested). Below is a summary of the comments and adjustments made to the manuscript:

1: Place greater emphasis on novelty of the wood-based age model relative to other age models (e.g., planktic foram 14C).

We adjusted the title of the manuscript (as suggested) to, "Extreme lowering of

deglacial seawater radiocarbon recorded by both epifaunal and infaunal benthic foraminifera in a wood-dated sediment core". We also added new text examining the usefulness and application of the wood-based age model. Also, we compared the usefulness of our wood 14C ages relative to work published after our manuscript was submitted to Climate of the Past (Zhao and Keigwin, 2018). We also added all the accepted and rejected wood 14C ages to an available figure (see attached).

2. Macrofauna consumption is not clear.

We added text and a new figure (now "Figure 6"; see attached) to better explain the apparent macrofaunal disturbance we observed in our sediment core.

3. Elaborate on the stratigraphic correlation of the two sediment cores based on color.

Photos of the sediment cores were not as useful as some unpublished XRF measurements we had, so we decided to use the ratio of Ca measured on our sediment cores (using XRF and normalized to terrestrial contribution by dividing by Al) to illustrate the rapid change in sedimentary composition at both sites. This new data is meant to be used as an estimate for the abundance of calcium carbonate microfossils, which sharply decrease (just as color darkens) during the early Holocene. Even better, we were able to add this new data to an existing figure (see attached).

4. Accounting for bioturbation and its impact on benthic foram 14C.

Old and new text in the first paragraph of the Discussion section explicitly states that bioturbation cannot explain the extreme deglacial lowering of these (and likely other) benthic foraminifera $\Delta$14C. We did not elaborate on the "off-maxima" foraminifera abundance 14C values, as suggested, because this work is beyond the scope of the study in hand, but is the subject of a future manuscript. We also softened text that described the $\Delta$14C record as lacking the "W" shape of earlier work.

5. Comments on interspecies 14C age offsets.

We clarified some text that was confusing, making sure to state that while the interspecies 14C age differences were not significantly offset on average, the standard deviation between these (on and off the foram abundance maxima) is large. We also made large changes to Figure 3 (see attached Figure 3C) to enable the reader to see the interspecies differences in both 14C age and $\Delta$14C.

6. Details about diagenesis.

We added new text and removed old text, as suggested. Good comments.

7. Additional discussion of buffering / appropriate referencing for this discussion.

We complied with all requests.

Line-by-line comments.

We complied with all requests or adjusted text in ways to match the request in all but one instance. This instance was the suggestion that we discuss the locations of the records showing and not showing the extreme lowering of $\Delta$14C during the deglaciation. We feel this is beyond the scope of the current manuscript.
* * *
[Figure]

**Fig. 1.** New "Figure 6": the macrofaunal disturbance

[Figure]

**Fig. 2.** New Figure 3, with color-coded species 14C and D14C

---

## Referee Comment (RC2) · Anonymous Referee #2 · 24 Sep 2018

Marchitto et al 2007 record of anomalous deglacial radiocarbon depletions (as recorded in infaunal benthic foraminifera) in the subtropical eastern North Pacific (at intermediate depths) set up an exciting debate regarding the Pacific Ocean ventilation and its role in deglacial atmospheric $CO_2$ rise. While some subsequent studies provided support for the reliability and the interpretation of Marchitto et al 2007, many studies raised critics to it. These critics relate to the reconstructions (chronology imperfections) and the interpretation (the source and the spatial extent of the recorded radiocarbon anomalies). In addition, bioturbation, diagenetic alterations and reworking are potential biases for any foraminiferal ventilation record. Rafter et al by using a different (and novel) approach for establishing the age model and dating monospecific benthic foraminifera (one epifaunal and three infaunal) from nearby records show similar deglacial large radiocarbon depletions providing a strong support to the fidelity of Marchitto et al 2007 record (while the source of radiocarbon anomalies remains an open and puzzling question). This important contribution largely rules out that the reconstructed deglacial radiocarbon depletions in Baja California are age model or foraminiferal artefacts. The paper is clearly written and the data are well presented and reasonably interpreted. I look forward to seeing it published after moderate, though critical, revisions.

First, the authors need to elaborate on their approach for constructing the age model, both advantages and complications. Given that the marine deposition of wood is very fast, the host sediment age = wood age – the time from wood growth to marine deposition. The premise is that the time from wood growth to marine deposition is more or less negligible. Are there any criteria that could be used to test that the dated wood have not been stored for long at land (i.e., the time from wood growth to marine deposition can be ignored)? For example, Zhue and Keigwin (2018, Nature Communications) used the presence of bark layers as a criterion that their wood samples are not redeposited old remains. A check to whether the wood age is younger than the age of co-deposited benthic foraminifera is employed to test the reliability of wood ages and accordingly 5 wood dates were omitted (as they showed older ages). Then, the 'test-passed' wood ages should be taken as the oldest possible ages for the host sediments (and reconstructed ventilation ages are the minimum possible ages) as the authors stated (page 6, line 25). While this issue does not affect the overall conclusion of this paper in terms of the validation of Marchitto et al record, the authors used the observation that their ventilation age estimates are not way older than those in Marchitto 2007 to partly rule out the effect of sedimentary redeposition (page 10, line 18) and diagenesis (Page 11, line 29). I think some consistency is needed here. What foram species has been compared to the wood; is it P. ariminensis (described as the 'preferred one'), or the species with the youngest age from each sample? Can long period from wood growth to marine deposition be (partly) masked (when ages from coeval wood and foram are compared)

due to the generally decreasing atmospheric 14C during the time interval of interest?

Second, in some places in the paper one might get the impression that the inference that the extreme deglacial radiocarbon anomalies are not species-specific is applicable to everywhere e.g., the paper title (the study area should be stated there); page 3, line 19; page 10, line, 29, referring to Thornalley et al 2011; the first paragraph of the 'conclusions' section in general e.g., mentioning the North Atlantic (page 12, line 5 ). As the authors stated in other places this is not the case e.g., the deglcial extreme aging from the subpolar North Atlantic and the southern Norwegian Sea seems to be Pyrgo-specific, away from the fact it's not yet clear whether the interspecies age differences from the these areas are due to short-term hydrographic changes or foram issues (Ezat et al., 2017).

Third, what do the authors think about the interspecies benthic 14C age differences in their records? Do they reflect real hydrographic changes? Can they be explained in the light of species abundance data (bioturbation effects)? I think these interspecies 14C age differences are the meant by '….-with important caveats-…..' in the abstract (line 12), and so they deserve more attention in the discussion. While the average values from abundance maxima for P. armininesis, U. peregrine, T. bardyi (but not Bolivina spp.) are similar, the range of variability is still large and deserves more attention in the discussion. Can d13C, d18O records of these different benthic species be helpful here (if they are measured)?

Fourth, the higher values that form the middle of the 'W' shaped anomaly in Marchitto et al 2007 (from 14.4 to 13.6 ka) is based on 1 mixed, 2 Uvigerina and 1 bolivina dates. The adopted explanation here to explain the absence of the middle of 'W' in the Gulf records would have been plausible if the middle of 'W' is based on T. bradyi, but this is not the case. Can the lower sedimentation rates in the Gulf records be the reason for the absence of this feature in the 'Gulf' records? Notably, however, the highest delta 14C value in this part is based on mixed benthic (so, it would be interesting to check with the authors of Marchitto et al 2007 if this mixed sample includes T. bradyi in a

significant amount (if 'detailed' picking notes were taken for this sample).

Fifth, I suggest plotting the isotope data from Lindsay et al 2015, 2016 in Figure 6, which I think will make the comparisons between 'Undercurrent' and 'Gulf' records clearer to the readers (page 9, line 15). Related to this, can d18O records be used to align the 'Undercurrent' and 'Gulf' records? If this is feasible, you may put the 'Undercurrent' records on your wood age model and calculate the 'Undercurrent' ventilation ages accordingly?

Minor comments

- Page 4, line 6, what size fraction used for foraminifera abundance? How is the error in abundance (as shown in Figure 4) calculated?

- Page 3, line 1, it's not clear for me what the sentence about 'planktic foraminifera' means. I think people usually try to account, though very difficult sometimes, for the vertical migration of planktic foraminifera by assigning an average (or a range of) calcification depth representative for the whole life period based on modern observations (e.g., from plankton tow, surface sample studies) (e.g., Sarnthein and Werner, 2017, Marine Micropaleontology).

- Page 4, line 27, for 0.7mg and 0.1 mg, I think those values are for carbon mass. Please specify to avoid confusion with carbonate weights.

- Page 5, line 7, 'sediments' or 'sediment cores' instead of 'sediment'?

- Page 5, line 8, '30-to12 kyr' instead of '30,000-to12,000 kyr', or change 'kyr' to 'years'?

- Page 8, line 15, as the Holocene is just tuned to be equal to modern (in terms of intermediate ventilation age), I think the deglacial values should be just compared to modern.

- Page 11, line 18, it's mentioned (in page 4, line 18) that tests with and without the HCl pretreatment yielded identical results for the investigated records.

[Figure]

- Page 12, line 15, '2015' is repeated.

- Figure 5e (or any other figure where all species-based ventilation ages are shown), why not giving each core a different colour and each species a different symbol? And if it's not going to be very messy, this can be done for Marchitto et al 2007 record as their supplemental table 1 includes the required information to do so?

———————————————

---

## Referee Comment (RC3) · Anonymous Referee #3 · 25 Sep 2018

Rafter et al present a very useful new deglacial 14C data set from the Gulf of California, which investigates species offsets in benthic forams. It provides new insight into the possible causes for and processes that could adversely impact 14C records in this region and globally. I concur with the points raised by Tom Marchitto in his thorough review and welcome the updated figures that enable a clearer picture of the raw data to be gained. I think the discussion is appropriate and although it doesn't solve the problem of the cause of the deglacial (and in some cases glacial) low D14C events, it certainly provides enough useful evidence to warrant publication. Other than points already raised by other reviewers, I only have a couple of minor suggestions:

In reference to line 10, referring to existing glacial-interglacial mono-specific epibenthic 14C dates: perhaps worth noting that some epibenthic 14C dates were published by Voelker et al 1998 (Radiocarbon) although these are only for the glacial; and Thornalley et al 2015 (Science) published monospecific C. wuellerstorfi 14C dates from the deep Norwegian Sea for both the (de)glacial and Holocene.

Building on this point and in reference to page 6, line 11, I think the wording could be changed because, as currently stated, the reader might take away the impression that the old ages reported in the subpolar North Atlantic are only related to samples including pyrgo. Yet this would be an incorrect impression because old deglacial ages are also reported in the deep Nordic Seas (strictly speaking, part of the North Atlantic) on monospecific C. wuellerstorfi samples. This is a useful point to incorporate in the work here because it suggests that extreme 14C depletions are not limited to just pyrgo or infaunal species, but have also been reported for C. wuellerstorfi - the benthic foram of choice for stable carbon isotope analysis, and which in this case (supported by pyrgo d13C) do not indicate the presence of low d13C waters causing the D14C depletions.

I agree with Marchitto that the text should be altered so that the records from around the globe are not necessarily conflated as representing one common signal. Although it is possible that a similar phenomena such as hydrothermal-volcanic fluids may be causing these signals (notwithstanding the issues of the d13C signature and the required seawater buffering needed), the different regions also have different oceanographic histories. For example the timing of old ages reported in the deep Southern Ocean and Pacific (Skinner et al 2010; Sykes et al 2000) and deep Nordic Seas (Thornalley et al 2015) is different to the intermediate depth sites and plausibly are consistent with the concept of isolated deep ocean reservoirs releasing their aged water in the deglacial (of course volcanic degassing/hydrothermal fluids may have helped contribute to the the extreme aging recorded).

---

## Author Comment (AC2) · 9 Oct 2018

Our thanks to Referee 2 for your comments. We adjusted our manuscript to all comments. The comments are listed numerically below, with our actions immediately following.

Comment 1.1. Can the authors elaborate on constructing of age model using wood. What is the criteria for acceptance?

Our test for accepting the 14C age of wood for our age model is that this age had to be older than ALL foraminifera in the same sample depth. This includes planktic 14C

ages, which were measured for all samples (but are not shown in this manuscript). We adjusted the text of the manuscript to reflect this clarification. While we also recovered wood that passed the "intact bark test" of Zhao and Keigwin (2018) (see Figure 3), we find that our quantitative 14C age test is appropriate.

Comment 1.2. Should clarify the potential errors in assuming quick deposition of wood in sediment. For example, ventilation age estimates (∆14C) might be older than the Marchitto (2007) and Lindsay (2016) results because of this assumption. Also, potential for "masking" of the wood 14C age difference caused by secular changes in atmospheric 14C during deposition.

We have adjusted the text in the Methods and Results sections to clarify the implications behind our assumption of a "contemporary deposition" of wood alongside our microfossils in Gulf of California sediment.

Here is preliminary adjusted text from the Methods section:

In light of this unusual application of calibrated 14C ages on wood in a marine setting, it is important to understand the potential errors. We assigned all calibrated wood ages a ±100 year uncertainty added in quadrature to the measurement and calibration error to account for possible lag in seafloor deposition. Note that the asymmetry of any errors associated with assuming contemporary growth of wood and foraminifera must be considered: if we underestimate the time from wood growth to sediment deposition, the actual calendar age of the sediment would be younger than the calendar age given in this study; hence foram ∆14C values would be even lower than the large depletions shown here (see equation 1 and Results). For example, it is possible that a longer-than-expected time period between wood growth and sediment deposition could be "masked" by declining atmospheric 14C concentrations (Figure 1), allowing the wood 14C age to pass our test for inclusion in the age model. This error would adjust our benthic foraminifera ∆14C values to lower values than reported below.

Here is the preliminary adjusted text from the Results section:

A wood 14C age-constrained age model has only been applied twice before (Broecker, 2004; Zhao and Keigwin, 2018) and it is worth quantifying the suitability of this approach in our cores. First, we applied a quantitative test: the wood 14C age must be older than all coexisting foraminifera 14C ages. This test included planktic foraminifera measurements that will be discussed in a following manuscript. The difference between benthic foraminifera and wood 14C ages is illustrative of the effectiveness of this test. The difference between the 14C age of benthic foraminifera (P. ariminensis and U. peregrina) and coexisting, wood that passed our test is 2346±1599 years (n=14) and 2309±1063 years (n=14), respectively (Table 2). Only comparing wood with foraminifera abundance maxima gives a 14C age difference of 3353±1957 years (P. ariminensis; maximum of 5815 years, minimum of 1077 years, n=6) and 2697±1117 years (U. peregrina; maximum of 4145 years, minimum of 1480 years, n=6). These values are consistent with bottom water at our core sites that are near or older than the modern, pre-bomb seawater - atmosphere 14C age difference of 1240 years (see above). Given these results, we argue that our test for excluding wood 14C ages is appropriate, but that unlikely circumstances may have existed that could hide the timescale of deposition. In the event of a longer-than-expected time between wood growth and deposition in the sediment, the calendar age would be biased to younger ages, making benthic foraminifera Δ14C values even more depleted than calculated (Figure 5).

Comment 2. "In some places in the paper one might get the impression that the inference that the extreme deglacial radiocarbon anomalies are not species-specific is applicable to everywhere e.g., the paper title". The rest of this comment is unclear to us, but seems to suggest that a full deglacial record of epifaunal mono-species 14C ages already exists (Thornalley et al. 2011).

We are unaware of another study that produces a full record of epifaunal mono-species 14C measurements. Accordingly, we argue the title is appropriate for this study. Addressing the last part of the comment above, the Thornalley et al. 2011 paper has a

section titled,

"Table S1. Benthic 14C dates. Mixed benthic foraminifera were picked from the >212mm fraction and were centred on local abundance maxima. Where samples from several adjacent 1 cm were combined, the assigned age is the weighted mean age of the sample period based on the mass of material from each 1 cm interval. Cibicidoides d13C from ref. S5."

In the case where the reviewer mistook the Thornalley et al. 2011 for Thornalley et al. 2015 (yet another Science paper), the appropriate quote from that study is:

"Mixed benthic ventilation ages. It was not always possible to obtain single genera benthic foraminifera ventilation ages, and therefore, to obtain a more complete deglacial record of ventilation changes, we had to use measurement made on mixed benthics."

Unless we have misunderstood the comment (our apologies if we have), the text above clearly states that mixed benthic species had to be used to obtain a full deglacial benthic foraminifera 14C record from either Thornalley paper.

With regards to a new title for our study, we have a new title based on comments from Reviewer #1 (to highlight the rare use of wood to constrain our age model), which we think this is an appropriate addition to an already overlong title. The new title is:

"Extreme lowering of deglacial seawater radiocarbon recorded by both epifaunal and infaunal benthic foraminifera in a wood-dated sediment core".

We do not find this title to be inaccurate or misleading.

Comment 3. What do the authors think about interspecies benthic 14C age differences in their records? Could stable isotope measurements help to explain these differences?

Our 14C measurements of four benthic foraminifera species are the first of their kind and we hope that our comparisons of the species' values both on and off abundance maxima are helpful for future applications of this widespread technique for reconstructing seawater 14C.

We understand the enthusiasm to further explore the difference between species 14C. New stable isotope measurements would be useful in this regard, but we find this important subject is outside of the scope of this study and rightly deserves a stand-alone study. Because these comments are public, we would like to advertise that we are looking for collaborators to further investigate the inter-species relationships with new stable isotope measurements and modeling.

Comment 4.1. The 'W' shape in the Marchitto (2007) study should not be dismissed so readily.

This echoes a similar comment from Tom Marchitto himself (Reviewer #1) and we have adjusted our text as such (underlined text):

It is possible that this and other some smaller-scale features of a mixed benthic △14C record reflect the bias of a particular species and/or the influence of bioturbation in our lower sedimentation rate sites.

Comment 4.2 The higher values of the 'W' shape in Marchitto et al. (2007) are based on 1 mixed, 2 Uvigerina spp., and 1 Bolivina spp. dates. Therefore, the explanation proffered in our manuscript (potential inclusion of T. bradyi) is not the case.

Unless our datasets are different, we politely disagree that the 'W' shape only contains 1 mixed species 14C date. The attached figure shows that the highest points of the 'W' shape are 7/9 mixed benthic measurements, including 2 of the 3 middle-high point, Bolivina spp. measurements make up the other 2 measurements. The bottom part of the 'W' and much of the lower values for the rest of the record are Uvigerina species. Therefore, we stand by our language that the 'W' shape is "not obviously reproduced by any of the 4 mono-species benthic foraminifera △14C."

Comment 5.1 Compare Undercurrent and Gulf benthic foram d18O.

This comparison would be between P. ariminensis (epifaunal) and Uvigerina spp. (infaunal), which comes with some caveats. The argument against including this additional data is that it clutters the figure, but we are willing to add this to our figure if the Editor feels it adds to the study.

Comment 5.2 Synchronize Pacific Margin site results (Marchitto and Lindsay studies) and calculate Undercurrent ventilation ages.

This is another good suggestion for future work and are open to collaborations on this and other subjects.

Minor Comments: 1. size fraction for abundance? This was also commented on by Reviewer 1 and added to the manuscript. It was >150 um.

2. Unclear why variable depths of planktic foraminifera calcification is an example of an "assumption about the reliability of the foraminifera archive" (our words). Studies often use modern observations to establish calcification depths. We have changed our text to read, "For example, an important assumption when using planktic foraminifera is that the depth of calcification does not vary based on modern observations"

3. Specify units for carbon. This was also noted by prior reviewer. Has been changed.

4. Change core to sediment cores Done.

5. Change kyr to yr. Done.

6. Holocene 14C ages are tuned to modern—use modern. Similar comment to Reviewer 1. Changed text to this: The shallowest and therefore most recent benthic foraminifera $\triangle$14C are roughly equal to modern DIC $\triangle$14C measurements at the depth of the cores of -173‰ (Key et al., 2004).

7. Detail the previously stated tests for removing authigenic carbonates. This has been changed in the text. Thank you.

8. Figure 5E. Add symbols to signify species. Done.

[Figure]

**Fig. 1.** Atmosphere and benthic foraminifera D14C (mixed, Uvi spp., and Bolivina spp.) from Marchitto et al. (2007) and Lindsay et al., (2016)

[Figure]

---

## Author Comment (AC3) · 9 Oct 2018

Response to Referee 3's review by Patrick Rafter (on behalf of all authors).

Our thanks for these useful comments and suggestions including commenting on the revised figures only recently uploaded for this conversation.

Comments 1 & 2. Note that there are other epibenthic 14C dates from the glacial, deglacial, and interglacial periods (refers to page 6, line 11 in original manuscript). Mistakenly implies that the inclusion of Pyrgo spp. 14C measurements impact all mixed benthic measurements.

This comment is echoed by Reviewer 2 and we have clarified text accordingly. Our intention is to state that our record is the only epifaunal benthic foraminifera 14C record that is *continuous* across the deglaciation. We have adjusted the text to:

A further complication to published benthic foraminifera △14C observations is that both the epifaunal and infaunal species are typically rare in sediments, leading to the common use of mixed benthic species. The mixed species approach has led, in some rare cases, to anomalously low △14C values / old 14C ages by inclusion of anomalously depleted 14C Pyrgo spp. (Magana et al., 2010)—an anomaly that may not be a global phenomenon (Thornalley et al., 2015). While mono-species epifaunal benthic foraminifera 14C measurements exist (Thornalley et al., 2011, 2015; Voelker et al., 1998), we are unaware of any continuous glacial-interglacial records of mono-species epifaunal foraminifera 14C content.

Comment 3. The depleted benthic foraminifera 14C records from around the globe should not be conflated (same comment as Reviewer 1 / T. Marchitto who referred to a statement in the first sentence of the Conclusions).

We adjusted the text in the conclusions to more accurately reflect the location of our study and the differences in timing between the records.

5.0 Conclusions If the extreme deglacial depletion of benthic foraminifera △14C at these northeastern Pacific sites cannot be explained by species or habitat bias, bioturbation, or poor age model control, the remaining explanation is that they reflect a change in seawater DIC △14C. The evidence in support of depleted seawater 14C content during the deglaciation (although often with different timing) includes deep-sea coral △14C measurements in both the Southern Ocean and North Atlantic (Adkins et al., 1998; Burke and Robinson, 2012; Chen et al., 2015; Robinson et al., 2005), which are often on rocky seamounts, have excellent age model control, and should not be influenced by the same diagenetic processes.

A leading candidate among the potential explanations for these and other intermediate

depth records (Bryan et al., 2010) is the deep-sea sequestration and flushing of carbon through the intermediate depth ocean (Basak et al., 2010; Du et al., 2018; Lindsay et al., 2016; Marchitto et al., 2007). This interpretation is plausibly consistent with 14C records from the deep Southern Ocean (Barker et al., 2010; Skinner et al., 2010) and deep Nordic Seas (Thornalley et al., 2015). However, a box model by (Hain et al., 2011) suggests that matching the observed $\Delta$14C depletions in the intermediate depth, Northern Hemisphere sites requires unrealistic changes in ocean chemistry (e.g., lower surface ocean alkalinity) and ocean dynamics (i.e., mixing).

---

## Author Response (AR1)

Editor Decision and Response by Authors (primarily Patrick Rafter)

In preparing your revised manuscript, I would request that you please highlight the changes that have been made.

Response: Please find our manuscript—with text updated based on these comments—as both a PDF (with tracked changes highlighted) and a MS Word document (also with tracked changes). The latter should provide the necessary text for publication, assuming our changes are acceptable. We re-state the editor's "additional points" and provide a response to each below.

1. Given the emphasis on 'monospecificity' in this study, I think it is important that your figures illustrating the new benthic foraminiferal radiocarbon data (Fig. 5 and 6) show four curves in each instance (one for each species 'category' shown in Figure 5). Indeed, I think that Figure 5E would also be more informative if the different records from each 'species category' were plotted together with different symbols/lines, instead of a single connected line). This suggestion was made by Reviewer 1 and was not taken up in your response, despite the very useful addition of differentiated benthic dates to Figure 2 (new Figure 3). The further suggestion of Reviewer 2 to also illustrate species used in the CU record with different symbols is also a good one. The interspecies offsets remain a major unresolved issue that is worth highlighting.

Response: The updated manuscript applies the approach from Figure 2 (new Figure 3) to illustrate the different species' values in Figures 5E and 7. We also updated the California Undercurrent site to illustrate the different species' values (data from Marchitto et al. 2007 and Lindsay et al. 2015). These new figures are a more clear visualization of the species differences. We did not apply separate lines connecting each species' values because this would result in a total of 6 overlapping lines (California Undercurrent data, 4 mono-species results, and mixed species results)— a true 'spaghetti'-like plot of lines. We felt that these plots are complicated enough with two lines overlapping plus symbols.

2. Again on Figure 5: it is not clear to me what the lines and symbols represent. Each figure has the CU record in red, and then another line with small black symbols, where some of these small symbols also correspond to large open symbols. Please clarify in the caption what the latter two are, as it currently only refers to one dataset for each subplot as far as I can tell.

Response: We have updated the caption. The larger symbols represent measurements at foraminifera abundance maxima.

3. On the subject of the wood-based age-model: the caution and clear explanation of the fact that these dates reflect maximum ages for a given sediment depth is a welcome inclusion, however, you emphasise in your response to Reviewer 2 that an important aspect of your 'quantitative test' is that the wood dates are

younger than both planktic and benthic dates. In this case it would be helpful to also add to Figure 2 (new Figure 3) all of the existing planktic dates, so that the reader can see how many of the wood dates can be 'tested' in this way. I think that it is important to acknowledge (as Zhao and Keigwin did not, but your manuscript arguably does better) that this is a situation where it is possible to know when the dates are 'really' wrong (e.g. by as much as the surface reservoir age), but not necessarily when they are right. I would note that a 100yr Gaussian uncertainty may not be the best way of dealing with this, given that we know the error in the wood ages is more likely to be biased in one direction (i.e. too old; ages are a maximum for a given depth). I appreciate that this is a tricky issue to address, beyond underlying the caveats, but the addition of the planktic dates to Figure 2 would help the reader see the extent to which they help act as 'tests' for the wood dates (and for how many).

Response: We have added the planktic 14C ages to Figure 3 and new text describing the potential errors when using wood 14C ages to constrain the age model (see paragraph beginning line 280).

4. On the conclusions: I concur with the reviewers that the consistency of the UC and Gulf records with others from around the word is overstated, and that perhaps some more careful wording of the conclusions is warranted, based on a consideration of a wider range of data and models For example, I think that the statement: "...a box model by (Hain et al., 2011) suggests that matching the observed  $\Delta 14C$  depletions in the intermediate depth, Northern Hemisphere sites requires unrealistic changes in ocean chemistry (e.g., lower surface ocean alkalinity) and ocean dynamics (i.e., mixing)" needs some further clarification (i.e. in terms of what exactly that model experiment consisted of, and was able to address strictly). Also, the statement that "evidence in support of depleted seawater 14C content during the deglaciation (although often with different timing) includes deep-sea coral  $\Delta 14C$  measurements in both the Southern Ocean and North Atlantic (Adkins et al., 1998; Burke and Robinson, 2012; Chen et al., 2015; Robinson et al., 2005)" is accurate but not quite apposite, since none of the coral data referenced show anything like the level of radiocarbon depletion exhibited in the Gulf and CU, and all show a step change around the Bolling-Allerod. This is a notable fact, as is the fact that the vast bulk of existing radiocarbon data from the LGM (i.e. prior to the putative 'flushing events') show that extreme radiocarbon depletion existed at a few sites, but was not at all widespread. 'From whence was this extremely old carbon flushed therefore?', one wonders.

Response 1: Beginning on Line 582, we have adjusted the text to follow your suggestions:

However, using an 18-box geochemical ocean-atmosphere model to simulate glacial-interglacial ocean circulation and carbon cycling, Hain et al., (2011) argue that matching the observed  $\Delta^{14}$ C depletions in the intermediate depth, Northern Hemisphere sites requires unrealistic changes in ocean chemistry

(e.g., lower surface ocean alkalinity) and ocean dynamics (i.e., mixing). Specifically, to appropriately "age" deep-sea 14C requires deep-sea anoxia, which is not observed. Furthermore, the release of this deep-sea 14C to intermediate depths would dissipate much quicker than the several thousand year anomaly shown in Figure 5E.

Response 2: We adjusted the text above and added text (see below) to more accurately and appropriately represent the deep-sea coral 14C (w.r.t. timing and magnitude):

If the extreme deglacial depletion of benthic foraminifera  $\Delta^{14}$ C at these northeastern Pacific sites cannot be explained by species or habitat bias, bioturbation, or poor age model control, the remaining explanation is that they reflect a change in seawater DIC  $\Delta^{14}$ C. Looking to other proxy systems, deep-sea coral  $\Delta^{14}$ C in the North Atlantic and Southern Ocean—archives with excellent age model control and different diagenetic influences—also display depleted deglacial  $\Delta^{14}$ C during the deglaciation (Adkins et al., 1998; Burke and Robinson, 2012; Chen et al., 2015; Robinson et al., 2005). However, the deep-sea coral  $\Delta^{14}$ C depletion have a different timing and are not as extreme as observed for the Gulf of California and California Undercurrent sites (Figure 5E).

**Extreme lowering of deglacial seawater radiocarbon recorded by both epifaunal and infaunal benthic foraminifera in a wood-dated**

**sediment core**

Authors: Patrick A. Rafter1\*, Juan-Carlos Herguera2, and John R. Southon1

**Affiliations:**

- 8 1Department of Earth System Science, University of California, Irvine, CA, USA
- 9 2Centro de Investigación Científica y Educación Superior de Ensenada 10
- 11 \*Correspondence to: prafter@uci.edu
- 12

3

4 5

6 7

- 13 Key Points
- 14 Carbon Cycle
- 15 Climate
- 16 Ice ages 17

**18 Abstract:**

- 19 For over a decade, oceanographers have debated the interpretation and reliability of
- 20 sediment microfossil records indicating extremely low seawater radiocarbon (14C)
- 21 during the last deglaciation—observations that suggest a major disruption in
- 22 marine carbon cycling coincident with rising atmospheric CO2 concentrations.
- Possible flaws in these records include poor age model controls, utilization of mixed,
- 24 infaunal foraminifera species, and bioturbation. We have addressed these concerns
- using a glacial-interglacial record of epifaunal benthic foraminifera 14C on an ideal
- 26 sedimentary age model (wood calibrated to atmosphere 14C). Our results affirm—
- 27 with important caveats—the fidelity of these microfossil archives and confirm
- 28 previous observations of highly depleted seawater 14C at intermediate depths in the
- 29 deglacial northeast Pacific.30

**31 1.0 Introduction**

[revised manuscript text omitted]

Patrick Rafter 9/20/2018 2:28 PM Deleted: 500

Patrick Rafter 9/20/2018 2:30 PM Deleted: equal to or Patrick Rafter 9/20/2018 2:32 PM Deleted: lower Patrick Rafter 9/20/2018 2:35 PM Deleted: during the deglaciation Patrick Rafter 9/20/2018 2:36 PM Deleted: , Patrick Rafter 9/20/2018 2:37 PM Deleted: able to propagate

97 not observed at all intermediate depth sites during the deglaciation (De Pol-Holz et 98 al., 2010; Rose et al., 2010). Furthermore, the extreme  $\Delta^{14}$ C lowering observed in 99 intermediate-depth benthic foraminifera during the deglaciation does not appear to 100 be quantitatively consistent with an isolated deep-sea reservoir (Hain et al., 2011). 101 102 The inconsistency of the available  $\Delta^{14}$ C records is compounded by assumptions 103 about the reliability of the foraminifera archive as a recorder of seawater DIC 14C. 104 For example, an important assumption when using planktic foraminifera is that the 105 depth of calcification does not vary based on modern observations (e.g., (Field, 106 2004)). The use of benthic foraminifera seemingly circumvents this problem, and 107 those that live at the sediment-water interface ("epifaunal") have been 108 demonstrated to record seawater carbon chemistry (Keigwin, 2002; Roach et al., 109 2013). However, the abundance of epifaunal benthic foraminifera is typically low 110 relative to benthic species that abide within the sediment ("infaunal"). Rather than 111 recording seawater 14C content directly, the infaunal species provide a record of 112 sediment pore water carbon chemistry, which may or may not reflect bottom water 113 conditions. 114 115 A further complication to published benthic foraminifera  $\Delta^{14}$ C observations is that 116 both the epifaunal and infaunal species are typically rare in sediments, leading to 117 the common use of mixed benthic species. The mixed species approach has led, in 118 some rare cases, to anomalously low  $\Delta^{14}$ C values / old 14C ages by inclusion of 119 anomalously depleted 14C Pyrgo spp. (Magana et al., 2010),—an anomaly that may 120 not be a global phenomenon (Thornalley et al., 2015). While mono-species epifaunal 121 benthic foraminifera 14C measurements exist (Thornalley et al., 2011, 2015; Voelker 122 et al., 1998), we are unaware of any continuous glacial-interglacial records of mono-123 species epifaunal foraminifera 14C content. (One study used mixed planispiral 124 species, whose morphology predicts an epifaunal habitat (Galbraith et al., 2007).) 125 An additional influence on benthic foraminifera  $\Delta^{14}$ C is bioturbation (Keigwin and 126 Guilderson, 2009), which is infrequently guantified, even though it can dramatically 127 affect the observed 14C age (Costa et al., 2018). The doubts raised by the above 128 complications are amplified by converting the benthic foraminifera 14C age to  $\Delta^{14}$ C. 129 which requires the user to assign a calendar age to the sediment. 130 131 Finally, constraining the age model of sediment cores typically relies upon several 132 assumptions. For example, planktic foraminifera 14C is commonly used to identify 133 the calendar age of sedimentary material, although this requires assumptions about 134 the depth habitat of the planktic foraminifera and the 'reservoir age' of the surface 135 waters (the offset between atmosphere and ocean 14C). Other means for 136 determining the calendar age involve tying temporal variability to other 137 paleoclimate/paleoceanographic records (Marchitto et al., 2007; Stott et al., 2009). 138 In rare instances, the 14C of wood from terrestrial plants provides a direct recording 139 of atmospheric 14C, which is well-dated and provides an excellent sedimentary age 140 model (Broecker, 2004; Zhao and Keigwin, 2018), although this work provides some 141 recommendations for utilizing this technique (see below). For our understanding of

**Patrick Rafter 9/27/2018 1:25 PM**

Patrick Rafter 9/20/2018 11:22 AM Deleted: , but Patrick Rafter 9/20/2018 11:22 AM Deleted: that live at the sediment-water interface ("epifaunal")

Patrick Rafter 9/27/2018 3:16 PM Deleted: which in at least Patrick Rafter 9/27/2018 3:31 PM Deleted: have been shown Patrick Rafter 9/27/2018 3:31 PM Deleted: give Patrick Rafter 9/27/2018 3:24 PM Deleted: (Magana et al., 2010) Patrick Rafter 9/27/2018 3:17 PM Deleted: i Patrick Rafter 9/27/2018 3:35 PM Deleted: n fact, Patrick Rafter 9/20/2018 2:40 PM Deleted: taken into consideration

- 155 past and future carbon cycling processes, it is essential that we thoroughly explore
- 156 these influences and build confidence in these sediment proxy records.
- 157

- 158 Here, we provide a test of the fidelity of the benthic foraminifera  $\Delta^{14}$ C proxy using 159 14C measurement of benthic foraminifera species from two sediment cores near th
  - 14C measurement of benthic foraminifera species from two sediment cores near the mouth of the Gulf of California (white diamond in Figure 2). These sediment cores
- are unusual in that both epifaunal and infaunal benthic foraminifera microfossils areplentiful and allow us a unique opportunity to test the fidelity of the benthic
- for a minifera  $\Delta^{14}$ C proxy. The for a miniferal abundance were quantified to account
- 164 for bioturbation and the age model is calibrated to the well-constrained
- 165 atmospheric 14C record (Reimer et al., 2013) via wood found alongside the
- 166 for a minifera. These cores (from hereon, the 'Gulf' sites) allow us to present glacial-
- 167 interglacial 14C measurements produced from 4 benthic foraminifera, including the
- 168 preferred epifaunal species *Planulina ariminensis* (Keigwin, 2002). The Gulf core
- 169 sites are bathed in the subsurface, northward flowing Mexican Coastal Current (MCC 170 in Figure 2), which are the source of the California Undercurrent (Gómez-Valdivia et
- in Figure 2), which are the source of the California Undercurrent (Gómez-Valdivia et
   al., 2015)—waters that also bathe the well known sites on the Pacific margin of Baja
- al., 2015)—waters that also bathe the well known sites on the Pacific margin of Baja
  California shown in Figure 1 (from hereon, the 'California Undercurrent' sites). This
- 172 cantorna shown in Figure 1 (noninercon, the cantorna of a cantorna shown in Figure 1 (noninercon, the cantorna of a cantorna shown in Figure 1 (noninercon, the cantorna of a cantorna of a cantorna shown in Figure 1 (noninercon, the cantorna of a can

[revised manuscript text omitted]
., 276 2013). 277 Patrick Rafter 9/20/2018 10:07 AM 278 2.5 Wood 14C age test Moved up [2]: Ages between these 279 Terrestrial plant life must have a younger 14C age / higher  $\Delta^{14}$ C than all constraints were estimated using BACON, as 280 contemporaneous foraminifera because of the air-sea difference in 14C content (e.g., was done for the LPAZ-21P cores. 281 see Figure 1) and we used this inherent 14C age difference to check for Patrick Rafter 9/27/2018 4:34 PM contemporary deposition of the wood and microfossils in Gulf sediments. Fourteen 282 Deleted: out of 20 microscopic wood fragment 14C ages passed the test and include one 283 Patrick Rafter 9/27/2018 9:02 AM 284 interval that may have been influenced by macrofauna consumption and excretion Deleted: Fifteen 285 has a wood 14C age that is younger than foraminifera (see below). 286 287 One wood measurement that spectacularly failed this test came from presumably 288 mid-to-late-Holocene sediment (i.e., <12-kyr BP aged sediments based on the depth 289 below seafloor). However this wood yielded a 14C age of >25-kyr (see upside down 290 triangles in Figure 3). We explain this remarkable 14C age difference as the erosion 291 and deposition of relict wood stored on land before washing to the Gulf during a 292 rain event. The other wood measurements that failed this test gave 14C ages typically 293 within measurement error or were  $\approx 1000^{14}$ C years older than for a minifera  $^{14}$ C age.
  - 6

300 In total, 5 out of 20 wood 14C measurements were older than foraminifera in our 301 sediment cores relative to 1 out of 26 wood 14C measurements by the only other 302 study with similar length age model (Zhao and Keigwin, 2018). This difference may 303 be because faster sedimentation rate of Zhao and Keigwin, (2018) (20-60 cm kyr-1) 304 leads to less bioturbation and a faster burial of the wood alongside foraminifera 305 microfossils. Otherwise, the difference in rejections could be explained by our 306 measurement of all wood, whereas (Zhao and Keigwin, 2018) only measured wood 307 that still retained bark.

308 309 In light of this unusual application of calibrated 14C ages on wood in a marine setting, 310 it is important to understand the potential errors. We assigned all calibrated wood ages a  $\pm 100$  year uncertainty added in quadrature to the measurement and 311 312 calibration error to account for possible lag in seafloor deposition. Note that the 313 asymmetry of any errors associated with assuming contemporary growth of wood 314 and foraminifera must be considered: if we underestimate the time from wood 315 growth to sediment deposition, the actual calendar age of the sediment would be 316 *younger* than the calendar age given in this study; hence foram  $\Delta^{14}$ C values would be 317 even *lower* than the large depletions shown here (see equation 1 and Results). 318 Additionally, it is possible that a longer-than-expected time period between wood 319 growth and sediment deposition could be "masked" by declining atmospheric 14C 320 concentrations (Figure 1), allowing the wood 14C age to pass our test for inclusion in 321 the age model. These different histories for the wood found in our sediment cores 322 would mean the calendar age is younger than we have assumed, adjusting our 323 benthic foraminifera  $\Delta^{14}$ C values to lower values than reported below. Given these 324 potential influences on a wood 14C age-constrained age model, the uncertainty 325 should primarily include the younger calendar age and not the ±100 year Gaussian 326 uncertainty we assume. However, without a more exhaustive statistical study of age 327 model errors when using wood, it is simpler and more conservative to utilize a 328 Gaussian age model error. 329

330 Given these potential errors, it is worth considering the modern 14C age difference 331 between seawater at the sediment-water interface and the atmosphere. A measurement of seawater DIC 14C age close to our core site and depth (at 22°N. 332 333 110°W at 598 m), gives a 14C age of 1240 years BP. Assuming that DIC at this depth 334 has not vet been seriously impacted by bomb 14C (Key et al., 2004) this would 335 predict a pre-bomb wood-to-benthic foraminifera 14C age difference of 1240 years 336 BP. This is consistent with our data presented below, where the 14C age difference 337 between concurrent wood and benthic foraminifera *P. ariminensis* and *U. peregrina* 338 varies between this and even larger 14C age differences (Table 2). 339

**340 **3.0 Results**

- 341 **3.1 Age model and sedimentation rates**
- 342 The old coretop age for the LPAZ-21P core (5.3-kyr BP) indicates a poor recovery of
- 343 the youngest sediments by the piston core, similar to nearby coring sites on the
- 344 Pacific margin (van Geen et al., 2003). The LPAZ-21PG gravity core calendar ages

**7**

**Patrick Rafter 9/27/2018 9:14 AM Moved (insertion) [3]**

**Patrick Rafter 9/27/2018 9:14 AM**

**Moved up [3]:** A measurement of seawater DIC 14C age close to our core site and depth (at 22°N, 110°W at 598 m), gives a 14C age of 1240 years BP. Assuming that DIC at this depth has not yet been seriously impacted by bomb 14C (Key et al., 2004) this would predict a pre-bomb wood-to-benthic foraminifera 14C age difference of 1240 years BP.

**Patrick Rafter 9/27/2018 9:32 AM**

[revised manuscript text omitted]
 Deleted: We find significant Patrick Rafter 9/20/2018 10:53 AM Deleted: with Patrick Rafter 9/27/2018 9:28 AM Deleted: Table 2 Patrick Rafter 9/20/2018 10:56 AM Deleted:

Patrick Rafter 9/20/2018 10:53 AM

Patrick Rafter 9/20/2018 10:54 AM Deleted: Comparing species only at abundance maxima Patrick Rafter 9/20/2018 10:54 AM Deleted: Patrick Rafter 9/20/2018 10:55 AM Deleted: considerably Patrick Rafter 9/20/2018 11:05 AM Deleted: , but t Patrick Rafter 9/27/2018 9:28 AM Deleted: Table 2

Patrick Rafter 9/27/2018 4:42 PM

- Patrick Rafter 9/27/2018 4:42 PM
- Deleted: ion

Patrick Rafter 9/27/2018 4:42 PM

Patrick Rafter 9/20/2018 2:57 PM

Patrick Rafter 9/27/2018 4:41 PM

[revised manuscript text omitted]

Patrick Rafter 9/20/2018 3:01 PM Deleted: where Patrick Rafter 9/20/2018 3:02 PM Deleted: at both sites Patrick Rafter 9/20/2018 3:03 PM Deleted: around 13-kyr BP Patrick Rafter 9/19/2018 2:13 PM Deleted: Figure 6

Patrick Rafter 9/27/2018 9:29 AM Deleted: Table 2

Patrick Rafter 9/20/2018 3:56 PM Deleted: which Patrick Rafter 9/20/2018 3:56 PM Deleted: was used

Patrick Rafter 10/8/2018 5:05 PM Deleted: " Patrick Rafter 10/8/2018 5:05 PM Deleted: "

Patrick Rafter 9/27/2018 4:42 PM Deleted: ( Patrick Rafter 9/27/2018 4:43 PM Deleted: (

- 585 older 14C ages, but a much larger reworked component (and hence much older
- 586 benthic foraminifera 14C ages) would logically be expected at the "upstream" Gulf
- 587 sites. In fact, sedimentary redeposition should be amplified at the lower
- 588 sedimentation rate Gulf site, but significantly lower benthic foraminifera  $\Delta^{14}$ C is not
- observed for any of the species at the Gulf sites.
- 590
- 591 These findings allow us to now focus our questions on two potential explanations
- 592 for the extreme depletions of benthic foraminifera  $\Delta^{14}$ C observed during the
- 593 deglaciation: (1) it is a diagenetic signal imparted onto both epifaunal and infaunal 594 foraminifera after burial or (2) it reflects a real change in seawater  $\Delta^{14}$ C during the
- for a for a fit or (2) it reflects a real change in seawater  $\Delta^{14}$ C during the deglaciation.
- 595 ( 596

**597 **4.2 Can diagenesis explain the low deglacial** $\Delta^{14}$ **C**?**

598 Investigating the potential for diagenetic alteration of benthic foraminifera  $\Delta^{14}$ C, we

- are not concerned about the newly observed coupling between carbonate
- dissolution and precipitation (Subhas et al., 2017), which only involves a few
- 601 monolayers of surface carbonate. Instead, producing the extreme  $\Delta^{14}$ C lowering
- 602 observed at Undercurrent and Gulf sites (Figure 5) and other sites around the globe
- 603 (Bryan et al., 2010; Stott et al., 2009; Thornalley et al., 2011) requires the 604 precipitation of depleted 14C on or within the foraminifera test is required.
- 605
- 606 This authigenic calcium carbonate formation and foraminifera 14C content has been
- 607 examined in several ways. For example, benthic foraminifera from the eastern
- 608 equatorial Pacific give, some of the lowest observed deglacial  $\Delta^{14}$ C values (-609‰),
- 609 but Scanning Electron Microscope images show no authigenic carbonate on benthic
- 610 or planktic foraminifera (Stott et al., 2009). Calcium carbonate overgrowth (via the
- 611 conversion of CaCO3 to CaSO4 (gypsum)) was observed in Santa Barbara Basin
- sediments (Magana et al., 2010), but would not influence the 14C content of the
- 613 microfossil. What's more, extreme 14C depletions of mixed benthic foraminifera
- 614 from this and other sites were found to be biased by *Pyrgo* spp., which are
- 615 inexplicably depleted in 14C (Ezat et al., 2017). Other work suggests *younger-than-*616 *expected* 14C ages from the precipitation of carbonate onto foraminifera tests after
- 617 core recovery (Skinner et al., 2010). Cook et al., (2011) observed anomalously low
- for a minifera  $\Delta^{14}$ C, high  $\delta^{18}$ O, and low  $\delta^{13}$ C was consistent with authigenic carbonate
- 610 precipitation from methane. Wycech et al., (2016) also compared the  $^{14}$ C ages of
- translucent and opaque mono-specific planktic foraminifera from the same
- 621 sediment horizons and found the opaque foraminifera (thought to contain
- 622 authigenic carbonate) had  $^{14}$ C ages more than 10,000 years older than the
- 623 translucent tests.
- 624

625 Neither the Gulf nor the Undercurrent site benthic foraminifera measurements

- 626 display the telltale signs of simultaneous  $\Delta^{14}$ C,  $\delta^{18}$ O, and  $\delta^{13}$ C anomalies seen by
- 627 Cook et al., (2011) (see Figure 7). What's more, the planktic  $\Delta^{14}$ C values from the
- 628 Undercurrent site do not show anomalous depletion during the deglaciation
- 629 (Lindsay et al., 2015), which is expected for post-depositional alteration / authigenic

Patrick Rafter 9/27/2018 4:43 PM Deleted: s Patrick Rafter 9/27/2018 4:43 PM Deleted: one

Patrick Rafter 9/20/2018 3:09 PM Deleted:

Patrick Rafter 9/27/2018 4:44 PM **Deleted:** Similarly,

Patrick Rafter 9/27/2018 4:44 PM Deleted: ( Patrick Rafter 9/19/2018 2:13 PM Deleted: Figure 6 Patrick Rafter 9/20/2018 5:19 PM Deleted: similarly

12

- 637 carbonate formation. It is possible that a completely different process of authigenic
- 638 carbonate formation is occurring in the subtropical eastern Pacific, but we cannot
- elaborate on what this mechanism might be. It is possible that authigenic carbonates

are removed from the foraminiferal test during the 10% acid leaching pre-treatment

641 at KCCAMS (see Methods), although selected pre-treatment tests did not

642 significantly alter the 14C ages. This pretreatment was not used in the Wycech et al.,

- 643 (2016) comparisons, but will be examined in our future studies.
- 644
- Finally, given the near identical deglacial  $\Delta^{14}$ C trends at the Undercurrent and Gulf
- sites despite very different sedimentation rates (20-30 cm kyr-1 at the Undercurrent
- 647 versus 1-to-5 cm kyr-1 at the Gulf; Figure 3) it would be surprising if the same
- 648 depleted  $\Delta^{14}$ C trends were of diagenetic origin. This is because a faster
- 649 sedimentation rate will decrease the potential for authigenic mineralization by
- decreasing the exposure time of the foraminifera. This reduction in exposure time
- 651 would apply to both the microfossil's exposure at the sediment-water interface and 652 at sediment depths favorable to authigenic carbonate precipitation. Thus, while the
- 652 at sediment depths favorable to authigenic carbonate precipitation. Thus, while the 653 potential influence of authigenic carbonate on the primary foraminifera record is an
- 654 important area of research that deserves further study, the similarity of the
- 655 Undercurrent and Gulf records argues against contamination from authigenic
- 656 carbonate precipitation as the major influence on these benthic foraminifera  $\Delta^{14}$ C 657 values.
- 658

**659 5.0 Conclusions**

660 If the extreme deglacial depletion of benthic foraminifera  $\Delta^{14}C$  at these northeastern 661 Pacific sites cannot be explained by species or habitat bias, bioturbation, or poor age 662 model control, the remaining explanation is that they reflect a change in seawater 663 DIC  $\Delta^{14}$ C. Looking to other proxy systems, deep-sea coral  $\Delta^{14}$ C in the North Atlantic 664 and Southern Ocean-archives with excellent age model control and different 665 diagenetic influences—also display depleted deglacial  $\Delta^{14}$ C during the deglaciation (Adkins et al., 1998; Burke and Robinson, 2012; Chen et al., 2015; Robinson et al., 666 **2005).** However, the deep-sea coral  $\Delta^{14}$ C depletion have a different timing and are 667 668 not as extreme as observed for the Gulf of California and California Undercurrent 669 sites (Figure 5E). 670

671 A leading candidate among the potential explanations for these and other

672 intermediate depth records (Bryan et al., 2010) is the deep-sea sequestration and

flushing of carbon through the intermediate depth ocean (Basak et al., 2010; Du et
al., 2018; Lindsay et al., 2016; Marchitto et al., 2007). This interpretation is plausibly

- 675 consistent with 14C records from only a few sites, such as the deep Southern Ocean
- 676 (Barker et al., 2010; Skinner et al., 2010) and deep Nordic Seas (Thornalley et al.,

677 2015). However, using an 18-box geochemical ocean-atmosphere model to simulate

- 678 glacial-interglacial ocean circulation and carbon cycling, Hain et al., (2011) argue
- 679 that matching the observed  $\Delta^{14}$ C depletions in the intermediate depth, Northern
- 680 Hemisphere sites requires unrealistic changes in ocean chemistry (e.g., lower
- 681 surface ocean alkalinity) and ocean dynamics (i.e., mixing). Specifically, to

13

Patrick Rafter 9/20/2018 5:17 PM **Deleted:** presumably

**Patrick Rafter 9/20/2018 1:20 PM Deleted: s appear to Patrick Rafter 9/20/2018 1:20 PM**

**Deleted:** be: (1) There was a near synchronous precipitation of 14C-depleted carbonate on to benthic foraminifera 'seeds' in different ocean basins (i.e., the eastern Pacific, southwest Pacific, Indian Ocean, and North Atlantic) or (2) The $\Delta^{14}$ C lowering**

Patrick Rafter 9/20/2018 1:20 PM Deleted: s

**Patrick Rafter 9/20/2018 1:20 PM**

**Patrick Rafter 11/12/2018 7:02 PM**

**Deleted:**

Patrick Rafter 11/12/2018 6:52 PM

**Patrick Rafter 11/12/2018 6:57 PM**

Patrick Rafter 9/20/2018 5:15 PM

- 702 appropriately "age" deep-sea 14C requires deep-sea anoxia, which is not observed. 703 Furthermore, the release of this deep-sea 14C to intermediate depths would 704 dissipate much quicker than the several thousand year anomaly shown in Figure 5E. 705 706 An alternative explanation involves the addition of 14C-depleted carbon via mid-707 ocean ridge (MOR) volcanism (Ronge et al., 2016), which is indirectly supported by 708 evidence for increased MOR activity (Lund, 2013; Middleton et al., 2016; Tolstoy, 709 2015), The locations and depths of the extreme benthic foraminifera  $\Delta^{14}$ C lowering 710 are also suggestive of a MOR influence, given their proximity to the East Pacific Rise 711 / Gulf of California (Marchitto et al., 2007; Ronge et al., 2016; Stott et al., 2009; this 712 study), the Red Sea (Bryan et al., 2010), and Mid-Atlantic Ridge (Thornalley et al., 713 2011). However, this hypothesis of enhanced carbon flux from seafloor volcanism 714 must also explain the many intermediate-depth sites that do not show anomalous 715 deglacial  $\Delta^{14}$ C depletions (Broecker & Clark, 2010; Cléroux et al., 2011; De Pol-Holz et al., 2010), Furthermore, this proposed carbon addition must have been associated 716 717 with an alkalinity addition, without which the increased seawater CO2 718 concentrations and therefore lower seawater pH would have caused a global-scale 719 carbonate dissolution event (Lindsay et al., 2016; Stott and Timmermann, 2011). 720 721 In summary, our work strongly suggests that at least for the Gulf of California and 722 adjacent Pacific sites, the foraminifera  $\Delta^{14}$ C proxy records real 14C changes in 723 deglacial intermediate depth seawater DIC, but the question of what caused those 724 changes remains open. Careful examination to confirm or disprove the fidelity of the 725 benthic foraminifera  $\Delta^{14}$ C on a case by case basis will be a critical part of building a 726 reliable body of data to identify the controls on glacial-interglacial marine carbon 727 cycling. 728 729 Acknowledgments: C. Bertrand, A. Hangsterfer (SIO Core Repository), H. Martinez, 730 N. Shammas, M. Ayad, M. Rudresh, A. De la Rosa, J. Troncoso, J. DeLine, J. Sanchez, C. 731 Manlapid, M. Chan, as well as T. Marchitto and two anonymous reviewers. 732 733 **References:** 734 Addison, J. A., Finney, B. P., Jaeger, J. M., Stoner, J. S., Norris, R. D. and Hangsterfer, A.: 735 Integrating satellite observations and modern climate measurements with the 736 recent sedimentary record: An example from Southeast Alaska: Modern SE Alaska 737 Fjord Sediment Records, J. Geophys. Res. Oceans, 118(7), 3444-3461, 738 doi:10.1002/jgrc.20243, 2013.
- Adkins, J. F., Cheng, H., Boyle, E. A., Druffel, E. R. M. and Edwards, L. R.: Deep-Sea
- Coral Evidence for Rapid Change in Ventilation of the Deep North Atlantic 15,400
- 741 Years Ago, Science, 280, 1998.
- Ahn, J. and Brook, E. J.: Siple Dome ice reveals two modes of millennial CO2 change
- 743 during the last ice age, Nat. Commun., 5(1), doi:10.1038/ncomms4723, 2014.

Patrick Rafter 9/27/2018 1:51 PM **Deleted:** (Lund, 2013; Middleton et al., 2016; Tolstoy, 2015, 2015)

Patrick Rafter 9/27/2018 4:20 PM Deleted: (e.g., Patrick Rafter 9/27/2018 4:20 PM Deleted: )

Patrick Rafter 9/27/2018 11:24 AM **Deleted:** and

[revised manuscript text omitted]

- 937

939